# Massively parallel RNA device engineering in mammalian cells with RNA-Seq

Joy S. Xiang[1], Matias Kaplan[1], Peter Dykstra [1], Michaela Hinks[1], Maureen McKeague[2,3] & Christina D. Smolke[1,4]*

Synthetic RNA-based genetic devices dynamically control a wide range of gene-regulatory processes across diverse cell types. However, the limited throughput of quantitative assays in mammalian cells has hindered fast iteration and interrogation of sequence space needed to identify new RNA devices. Here we report developing a quantitative, rapid and high-throughput mammalian cell-based RNA-Seq assay to efficiently engineer RNA devices. We identify new ribozyme-based RNA devices that respond to theophylline, hypoxanthine, cyclic-di-GMP, and folinic acid from libraries of ~22,700 sequences in total. The small molecule responsive devices exhibit low basal expression and high activation ratios, significantly expanding our toolset of highly functional ribozyme switches. The large datasets obtained further provide conserved sequence and structure motifs that may be used for rationally guided design. The RNA-Seq approach offers a generally applicable strategy for developing broad classes of RNA devices, thereby advancing the engineering of genetic devices for mammalian systems.

[1] Department of Bioengineering, 443 Via Ortega, MC 4245, Stanford University, Stanford, CA 94305, USA. [2] Department of Pharmacology and Therapeutics, McGill University, 3655 Prom. Sir-William-OslerMontrealQuebec H3G 1Y6, Canada. [3] Department of Chemistry, McGill University, 801 Sherbrooke Street WestMontrealQuebec H3A 0B8, Canada. [4] Chan Zuckerberg Biohub, San Francisco, CA 94158, USA. *email: csmolke@stanford.edu

Recent developments in mammalian synthetic biology have given rise to new tools and approaches for manipulating molecular interactions and regulating gene expression[1–6], and have been applied in optimizing efficacy while mitigating undesirable toxicity effects of molecular and cellular therapeutics[7–11]. RNA-based gene control devices can operate on the same time scale and subcellular location of numerous RNA-dependent cellular processes, and provide covalently linked *cis*-regulation of RNA molecules to minimize off-target effects. RNA therapeutics are poised to revolutionize medicine, due to their lower risk of genome integration than DNA therapies and of immunogenicity than antibody-based biologics[12]. Furthermore, RNA devices can modularly incorporate a variety of ligand-sensing aptamers for small molecule or protein dependent spatiotemporal control.

Ribozyme switches (also referred to as allosteric ribozymes or aptazymes) regulate diverse RNA processes, including RNA interference[13], CRISPR/Cas9 activity[14], and mRNA stability[15–18]. For post-transcriptional gene expression control in eukaryotic cells, a ribozyme switch is generally inserted into the 3′ untranslated region (3′ UTR) of a gene of interest, where ribozyme self-cleavage exposes cleaved ends to rapid exonuclease degradation[19]. Ligand binding at the aptamer domain inhibits ribozyme self-cleavage and stabilizes the transcript against exonuclease degradation, increasing mRNA and protein levels in a concentration-dependent manner (Fig. 1a). Ribozyme switches are advantageous over inducible expression systems requiring heterologous regulatory proteins, such as transcription factors, due to their small genetic footprint of ~100–200 nucleotides and independence from host regulatory proteins[20], which reduces the potential of immunogenicity.

Despite numerous proof-of-concept studies, synthetic ribozyme switches suffer from small dynamic ranges (up to ~3.5-fold in activating fluorescent reporter expression[21]) and are limited to a small, recurring set of aptamers that are functional in mammalian cells. Screening thousands of switch library variants by cell-free[22,23], FACS in yeast[24] or bacterial systems[25] can be performed with relative ease and has identified switches with improved sensitivities and dynamic ranges. However, differences between cell-free, microbial, and mammalian cell systems[26,27] can impact the self-cleavage rate, including differences in transcription rates[28] and intracellular Mg2+ concentration, or events downstream of the cleavage event, including differences in compartmentalization of transcription and translation, mRNA processing and degradation, and ligand permeability and toxicity, which may prevent switches from functioning across the systems. The throughput of methods for developing genetic devices in mammalian cells is limited to only tens to hundreds of variants, often requiring cloning and assaying individual sequences[17,21,29,30]. FACS-Seq in mammalian cells has been used to assay the gene-regulatory effect of translational initiation start site (TIS)[31] and internal ribosomal entry site (IRES) sequence variants[32], but the process involves handling large quantities of lentiviral and cell culture volumes, which proves cumbersome for rapid iterations through the engineering design-build-test cycles. Recently, an RNA-based sequencing method screened thousands of pistol ribozyme sequences in mammalian cells for self-cleavage activity[33]. However, the method relied on gel-based separation and extraction of cleaved and uncleaved RNA, which was prone to variability across experiments. The ribozyme libraries were also transcribed by a U6 promoter, which could give rise to differences in transcription, RNA stability, and degradation from RNA Pol II-based mRNA expression; as such the results may not be applicable for assaying the activity of ribozyme switches for gene expression control. High-throughput and scalable strategies are thus critically needed to facilitate the engineering of ligand-responsive switches in mammalian cells.

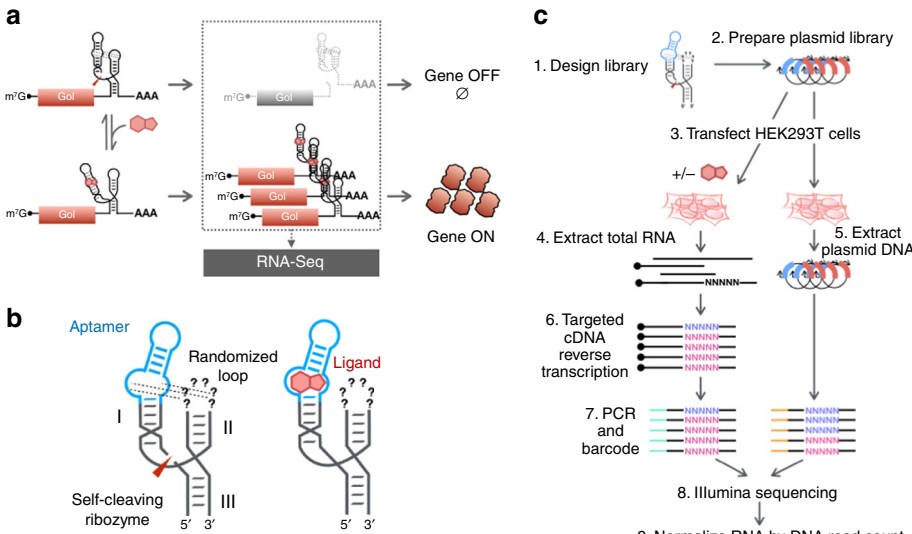

**Fig. 1** A high-throughput, quantitative pipeline for engineering ribozyme switches. **a** A schematic illustrating the mechanism by which ribozyme switches achieve conditional gene expression regulation. In the absence of ligand, the switch undergoes self-cleavage, destabilizing the transcript and resulting in low protein expression. Ligand binding to the switch interferes with the loop–loop tertiary interactions of the ribozyme to inhibit self-cleavage, in turn stabilizing the transcript and leading to increased protein expression. The differential mRNA levels resulting from ribozyme switch activity can be assayed with RNA-Seq. **b** Schematic of the architecture of a ribozyme switch. The switch comprises a ribozyme actuator domain (black) and a ligand-sensing aptamer domain (blue). In the illustrated library design, an aptamer is grafted onto stem I of the hammerhead ribozyme and 4–6 bases of stem loop II are randomized. The library is screened for loop sequences that interact with the integrated aptamer to enable ribozyme self-cleavage, while allowing ligand binding at the aptamer to disrupt the tertiary interactions and inhibit cleavage. **c** RNA-Seq workflow for measuring differential mRNA expression levels associated with a ribozyme switch library

We develop a quantitative, gel-free, and massively parallel RNA-Seq approach to simultaneously measure mRNA levels associated with tens of thousands of ribozyme switch library sequences in mammalian cells. From libraries incorporating diverse aptamers, we identify switches specific to theophylline, hypoxanthine, folinic acid, and cyclic di-GMP with low basal (i.e., in no ligand condition) expression levels of ~10%, and up to ninefold activation ratios. The RNA-Seq derived mRNA levels are predictive of protein expression levels as measured by FACS-Seq. Importantly, RNA-Seq is simpler, requiring off-the-shelf kits and only a week to obtain results compared with a month for FACS-Seq, and is nearly twice as accurate with a stronger $R^2$ correlation with validation assays. We further identify sequence and structural motifs that underlie the regulatory strength across different switch libraries, to derive a common, purine-rich motif for rapid, semi-rational design of new ribozyme switches, and inform future rational design efforts. We expect the RNA-Seq strategy to be applicable to engineering other classes of RNA devices to advance mammalian cell device engineering.

## Results

**RNA-Seq assays ribozyme switch libraries in mammalian cells.** Our ribozyme switch designs are conceptualized as a sensor domain (encoded by an aptamer) which is coupled to an actuator domain (encoded by a self-cleaving ribozyme) (Fig. 1b). We used the hammerhead ribozyme as the actuator domain, which requires tertiary interactions between nucleotides in stem loops I and II to stabilize the ribozyme structure for self-cleavage under physiological conditions[34,35]. In designing switch libraries, aptamers are integrated into one of two hammerhead ribozyme stem loops (loop I or loop II) and the sequence of the non-aptamer loop is randomized (Fig. 1b). A high-throughput functional assay is used to identify loop sequences that (1) form favorable tertiary interactions with the aptamer to support fast cleavage rates in the absence of ligand, and (2) allow ligand binding to disrupt these tertiary interactions and inhibit cleavage.

We developed a targeted short-read RNA-Seq approach to provide a quantitative, high-throughput readout of the activities of individual ribozyme switch library members, mapping sequence to transcript level (Fig. 1c, Supplementary Fig. 1). In the absence of the aptamer ligand a ribozyme switch is fast-cleaving and results in low basal mRNA and protein expression, whereas in the presence of ligand, ribozyme self-cleavage is inhibited to increase mRNA and protein expression. Therefore, the activity of a ribozyme switch can be measured by either transcript or protein levels for the target gene in the presence and absence of ligand.

We hypothesized that RNA read count normalized to DNA read count would reduce noise introduced by variations in DNA copy number due to limitations in evenly mixing library oligonucleotides in DNA synthesis. To examine variability in DNA copy number and transcript levels of each variant in an RNA device library independent of any self-cleavage, the RNA-Seq workflow was performed on a non-cleaving ribozyme library harboring a scrambled catalytic core and randomized loop II sequences (Supplementary Fig. 1, Supplementary Table 1). The library was cloned into the 3′ UTR of a *mCherry* expression construct in a plasmid vector (pCS4076) and transfected into HEK293T cells. The plasmid DNA libraries were prepared for sequencing analysis from the post-transfection plasmid library extracted from separately transfected cells. Quantifying DNA abundance post-transfection can account for additional sources of bias, such as low transfection efficiency and cell-to-cell variability in plasmid uptake. The RNA library was prepared for sequencing analysis by extraction of total RNA from transfected cells and subsequent reverse transcription to cDNA.

Deep sequencing for the above experiment showed strong replicate correlation for DNA and RNA read counts ($R^2 = 0.99$ and $R^2 = 0.99$, respectively, Supplementary Fig. 1) indicating that the abundance of individual library members are preserved. RNA and DNA read counts are also strongly correlated ($R^2 = 0.98$; Supplementary Fig. 1). During library construction, libraries were mixed with five control ribozymes (Supplementary Table 2) spanning a range of self-cleavage rates—the wild-type satellite RNA of the Tobacco Ringspot Virus (sTRSV) hammerhead ribozyme, three sTRSV-derived ribozymes with modified loops I and II sequences[24], and a non-cleaving mutated sTRSV ribozyme. The first four ribozymes undergo different extents of self-cleavage and were observed to deviate from a 1:1 RNA:DNA correlation (Supplementary Fig. 1). The normalized mRNA levels for the five control ribozymes strongly correlate ($R^2 = 0.95$ or 0.99) with qPCR measurements (Supplementary Fig. 2), whereas the unnormalized mRNA levels correlate less well ($R^2 = 0.89$ or 0.83). The normalized RNA levels of spiked-in ribozymes exhibit strong linear correlation with *mCherry* expression levels as measured by flow cytometry ($R^2 = 0.96$ or 0.99). To accurately measure ribozyme cleavage-dependent RNA variability, the variability in DNA abundance (e.g., from copy number variability) should thus be accounted for by normalizing RNA read count to DNA read count.

We applied the RNA-Seq assay in mammalian cells to analyze a ribozyme switch library responsive to theophylline. A library with the theophylline aptamer grafted onto stem I of the sTRSV hammerhead ribozyme and the opposing stem II loop consisting of 4–5 (N4–5) degenerate bases was designed (Fig. 2a) based on previous work[24] and tailored to a smaller sequence space expected to be enriched for fast-cleaving switches (1280 unique sequences). The switch library was cloned into the 3′ UTR of *mCherry* and transfected into HEK293T cells. Cells were incubated in 0 or 5 mM theophylline for 3 days, and total RNA was extracted, reverse transcribed, barcoded, and sequenced. The DNA plasmid library was extracted from separately transfected cells and similarly prepared for sequencing.

The RNA-Seq strategy quantitatively assayed every variant in the theophylline library to identify sequences exhibiting low mRNA expression in the absence of ligand and large dynamic ranges. By comparing normalized mRNA levels of library members at 0 and 5 mM theophylline, 148 sequences exhibited more than twofold increase in mRNA levels (designated as activation ratio >2; Fig. 2c). The maximum activation ratio observed is 6.8, with loop II sequence CAUAA (Supplementary Table 3). Of the 1280 unique sequences that were sequenced, 665 have a mean DNA sequence read coverage of >100 read count (empirically determined cutoff, see Supplementary Fig. 3) to give a replicate correlation of $R^2 = 0.80$ (Fig. 2b), and this subset was used in the above analysis. All cleaving sequences show some ligand-dependent increase in mRNA levels, where those with lower basal expression exhibit greater activation levels (Fig. 2c, inset). Thus, starting with library designs that exhibit greater bulk cleavage levels is important for generating switches with high activation ratios.

Sequences exhibiting high activation ratios and spanning a range of normalized mRNA levels were individually validated via flow cytometry for reporter gene expression via transient transfection in HEK293T cells. Switch candidates were verified to increase gene expression when induced by 1 mM and 5 mM theophylline, achieving up to ninefold activation at 5 mM theophylline (Fig. 2d). Three of the switches were further characterized to demonstrate a dose response relationship between relative fluorescence levels and theophylline

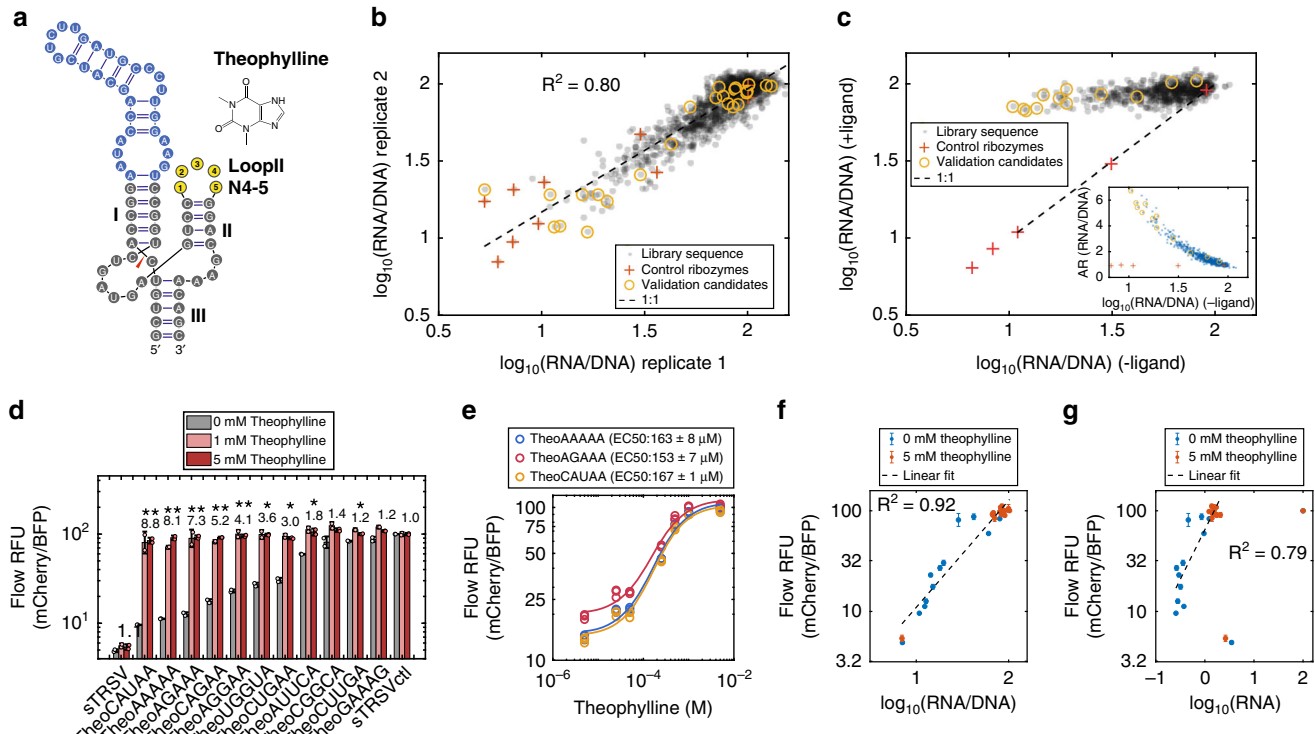

**Fig. 2** RNA-Seq assay for a theophylline ribozyme switch library. **a** Illustration of a theophylline ribozyme switch library design. **b** Normalized RNA read counts from replicate RNA-Seq experiments ($R^2 = 0.80$). Each dot represents a unique sequence with over 100 sequencing read counts in an experiment (665 sequences). Pluses are spiked-in control ribozymes; circles indicate sequences selected for validation. **c** Normalized RNA read counts from an RNA-Seq assay performed on a theophylline ribozyme switch library in the absence and presence of ligand. The inset shows the activation ratio (AR) as a function of normalized RNA read counts in the absence of ligand. −/+ Ligand conditions are 0/5 mM theophylline. **d** Flow cytometry analysis of individual members in the theophylline ribozyme switch library. Activation ratio (AR) of mCherry/BFP at the higher ligand concentration is indicated above each set of bars. sTRSV is the wild-type hammerhead ribozyme; sTRSVctl is a non-cleaving mutant of sTRSV; error bars indicate standard deviation of two biological replicates; asterisks indicate the Benjamini–Hochberg corrected $p$-values of switching significance between 0 and 5 mM theophylline, using $p$-values from unpaired, one-tailed $t$-test, $*p < 0.01$, $**p < 0.001$. Filled circles are individual replicate data points. **e** Dose response curves for three theophylline switches showing relative fluorescence values as a function of theophylline concentration. EC50 values are reported as mean ± standard deviation of three biological replicates. **f** Relative fluorescence values from flow cytometry analysis of individual sequences plotted against normalized RNA read counts from an RNA-Seq assay ($R^2 = 0.92$). Error bars indicate standard deviation of two biological replicates. **g** Relative fluorescence values from flow cytometry analysis of individual sequences plotted against unnormalized RNA read counts from an RNA-Seq assay ($R^2 = 0.78$). Error bars indicate standard deviation of two biological replicates. Outliers are spiked-in control ribozymes sTRSV and sTRSVctl. **a**, **c**, **d** All $R^2$ are based on $\log_{10}$ transformed values. Source data are available in the Source Data file

concentration with EC50 ranging from 153 to 167 μM (Fig. 2e). Thus, the RNA-Seq screen effectively identified the rare, high-performing switches from a large library.

The data indicate a strong linear relationship between normalized RNA count from the RNA-Seq assay and mean fluorescence of the individually validated theophylline switch candidates ($R^2 = 0.92$; Fig. 2f), where the correlation is weaker for RNA read count alone ($R^2 = 0.79$; Fig. 2g). Normalized RNA levels are comparable with spiked-in control sequences, which are subject to DNA quantification and mixing errors. While mRNA and protein levels (by RNA-Seq and flow cytometry, respectively) are correlated for the handful of validated switches (Fig. 2f), it is unknown whether there is library-wide agreement between mRNA and protein-level regulation, which may impact the accuracy of using RNA-Seq for engineering new ribozyme switches. Thus, we developed a mammalian cell FACS-Seq workflow to examine the protein expression regulatory activities of RNA device libraries.

## FACS-Seq shows correlated mRNA and protein levels in library. Mammalian cell FACS-Seq has been used to characterize

sequence-dependent effects of genetic elements on protein expression for tens of thousands of library sequences[31,32]. To link phenotype measurements of cellular fluorescence to the underlying genotype sequence, lentiviral library integration is typically used as a low MOI results in a large percentage of transduced cells undergoing at most one chromosomal integration. The workflow involves cloning a library into a lentiviral transfer plasmid, which is packaged into viral particles for transduction (Supplementary Fig. 4). The transduced cells are sorted into bins that span a range of fluorescence intensities, before genomic DNA extraction and PCR barcoding for Illumina sequencing. Sequencing read counts are used to reconstruct a histogram of fluorescence activity for each library sequence, which is fitted to a Gaussian distribution to estimate the mean fluorescence activity across a population of cells.

The FACS-Seq assay of the theophylline library, as used in RNA-Seq, was performed to characterize the mean fluorescence activity of 1202 unique library sequences having at least 100 NGS read coverage per sequence (based on the RNA-Seq cutoff) for each replicate and ligand condition (Supplementary Table 4). 1 mM theophylline was used instead of 5 mM to reduce toxicity to cells and maintain library coverage. To obtain the mean

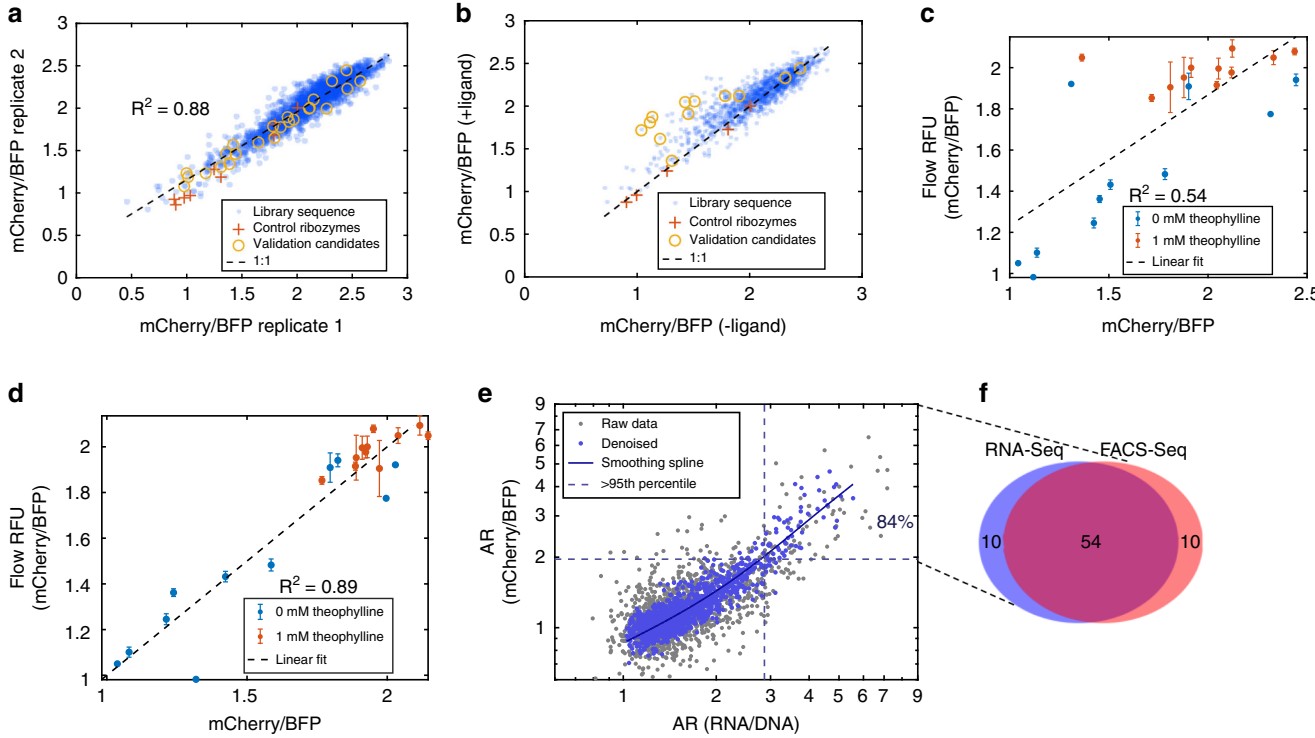

**Fig. 3** FACS-seq assay of a theophylline ribozyme switch library. **a** Relative fluorescence values from replicate FACS-Seq experiments ($R^2 = 0.87$) on a $\log_{10}$ scale. Each dot represents a unique sequence with over 100 sequencing read counts in an experiment (1202 sequences). Pluses are spiked-in control ribozymes; circles indicate sequences selected for validation. **b** Relative fluorescence values from a FACS-Seq assay performed on a theophylline ribozyme switch library in the absence and presence of ligand. −/+ Ligand conditions are 0/1 mM theophylline. **c** Relative fluorescence values from flow cytometry analysis of individual sequences plotted against relative fluorescence values from a FACS-seq assay ($R^2 = 0.55$). Error bars indicate standard deviation of two biological replicates. **d** Relative fluorescence values from flow cytometry analysis of individual sequences plotted against denoised relative fluorescence values from a FACS-seq assay ($R^2 = 0.89$). Error bars as in **c**. **e** Activation ratios of relative fluorescence values from FACS-seq plotted against activation ratios of normalized RNA read count from RNA-Seq. Raw data were denoised using a regression model from AutoML; from the denoised data both assays agree on ~84% of sequences with >95th percentile activation ratio. Dashed lines indicate the 95th percentile AR thresholds. **f** Venn diagram shows the number of sequences shared between RNA-Seq and FACS-Seq for sequences with >95th percentile activation ratio. **a**, **c**, **d** All $R^2$ are based on $\log_{10}$ transformed values. Source data are available in the Source Data file

mCherry/BFP ratio, the frequency of a given sequence across six bins was fit to a log-normal distribution by $X \sim N(\mu,\sigma)$, where $\mu$ is mean and $\sigma$ is standard deviation of each sequence across a population of cells. Replicate sequences demonstrate a linear correlation ($R^2 = 0.88$; Fig. 3a). Nineteen sequences exhibit significant differences in mean fluorescence in the presence and absence of ligand ($p$-value $< 1e{-}125$, or a Benjamini–Hochberg (BHFDR) adjusted $p < 1e{-}133$, by two-tailed unpaired $t$-test, chosen to minimize false positive "OFF" switches) with an activation ratio ranging from 2 to 3.8 (Fig. 3b; Supplementary Table 4). Mean fluorescence values of library members correlate between the FACS-Seq and flow cytometry validation results ($R^2 = 0.55$; Fig. 3c), though with higher noise than the RNA-Seq assay (see Supplementary Note 1).

We further denoised[36–38] the FACS-Seq data using an automated regression modeling algorithm[39], which improved the correlation with flow cytometry validation ($R^2 = 0.89$; Fig. 3d). Comparing the denoised results, there is a monotonically increasing relationship between the activation ratios in protein and mRNA levels and no distinct clusters of sequences deviate from the fitted spline curve (Fig. 3e). Basal levels of normalized mRNA and fluorescence levels also correlate (Supplementary Note 2; Supplementary Fig. 5). Fifty-four of 64 (84%) sequences are shared between FACS-Seq and RNA-Seq in the top 5th percentile of switches identified from each assay (as defined by

activation ratio levels) (Fig. 3e, Supplementary Fig. 6). This data, along with qPCR and flow cytometry validation of control ribozymes (Supplementary Fig. 2), support that regulating mRNA levels directly modulate protein expression. Taken together, RNA-Seq can directly predict the protein-level regulatory activity of ribozyme switches.

**RNA-Seq identifies xanthine, folinic acid, and c-di-GMP switches.** The availability of genetic switches responsive to ligands with low cytotoxicity and not antibiotics[40] is limited for applications in cell engineering and molecular therapeutics. While theophylline is an antitussive therapeutic and has been used in engineering diverse RNA devices, its applications are limited due to toxicity[41]; as such we explored several other aptamer ligands[41]. The xanthine aptamer exhibits a low micro-molar binding affinity to natural metabolites xanthine, guanine, and hypoxanthine[42]. The bacterial second messenger cyclic diguanylate monophosphate (cyclic di-GMP) may be employed as a vaccine adjuvant to elicit innate immune response in cell therapy[43,44], and the aptamer sequence can be extracted from the naturally occurring cyclic di-GMP-I and -II riboswitches. Finally, the aptamer[45,46] to the dextrorotatory isomer of folinic acid exhibits low nanomolar affinity to its ligand. While the levorotatory form of the folinic acid is the pharmacologically active

drug[47], folinic acid is composed of a racemic 1:1 mixture. Ribozyme switches to these ligands have been engineered in cell-free[48] or microbial systems[15,46], but they have not yet shown switching activity in mammalian cells. Thus, we applied the RNA-Seq approach to identify high-performing switches that respond to hypoxanthine, cyclic di-GMP, and folinic acid, to demonstrate the broad utility of our strategy to engineer new ligand-responsive switches.

To design the switch libraries, one of the stem loop sequences is replaced with the aptamer sequence, while the opposing loop is randomized with 5 degenerate bases (N5), similar to the theophylline library. When integrating the aptamer into the ribozyme stem loop, one or two base pairs in stem I or II is replaced with the terminal stem base pair of the aptamer to preserve the binding affinity ($K_D$) of the original aptamer domain (Supplementary Fig. 7)[49]. The aptamer-integrated libraries that preserved aptamer-only binding affinities were subsequently characterized for cleavage activity in an in vitro cotranscriptional RNA cleavage assay, where relative levels of cleaved and uncleaved RNA products were analyzed via polyacrylamide gel electrophoresis (PAGE). Only designs that met both criteria—aptamer domain binding affinity and ribozyme library bulk cleavage—were screened for cellular activities in RNA-Seq.

We next pre-screened the different library designs (Supplementary Fig. 7) to minimize library diversity, such that each library has coverage sufficient for quantitative analyses from a single MiSeq run format. The final library designs with xanthine, folinic acid, cyclic di-GMP-I, and cyclic di-GMP-II aptamers exhibit dissociation constants ($K_D$) of 17 μM, 390 nM, 750 nM, and 4.3 μM, respectively (Supplementary Fig. 7), and were verified to undergo self-cleavage in vitro (Supplementary Fig. 8). While the xanthine and folinic acid aptamer integrations required replacing one base pair in stem I with a base pair from the aptamer base stem, two base pair replacements were required for the cyclic di-GMP libraries. In the final library designs N5 and N6 randomized opposing stem loops were included to expand the sequence search space (Fig. 4a–c).

Final ribozyme switch libraries for the new aptamer–ligand pairs were characterized through RNA-Seq (Fig. 4d–f, Supplementary Table 3), along with spiked-in ribozymes as nonswitching controls. Assays were performed in HEK293T cells, except for the folinic acid ribozyme switch library, which was performed in HEK293T cells overexpressing the human folate transporter SLC46A1 (Supplementary Fig. 9). All unique sequences (5120) in each designed library were sequenced; however, only library members with more than 100 read count coverage were included in the analysis based on the cutoff set with the theophylline library (5073 xanthine library sequences, 4966 c-di-GMP-II library sequences, 1011 cyclic di-GMP-I library sequences, 5114 folinic acid library sequences). RNA-Seq measurements demonstrate good correlation between replicate experiments ($R^2 > 0.83$; Supplementary Fig. 10). Candidate switches exhibiting the lowest normalized RNA levels in the absence of ligand and highest activation ratios were selected for validation via flow cytometry analysis.

The RNA-Seq strategy generated xanthine ribozyme switches exhibiting low basal expression levels and high dynamic ranges of up to 5.6-fold (Supplementary Table 3). Eight library members were selected for individual validation via flow cytometry, and the data indicate good correlation between the RNA-Seq and flow cytometry analysis ($R^2 = 0.97$; Supplementary Fig. 11). The validated xanthine switch candidates exhibit activation ratios spanning 4.5–6.8-fold (mCherry/BFP ratio) and basal reporter protein expression levels as low as 12% that of sTRSVctl (Fig. 4j).

Analysis of the folinic acid library with RNA-Seq highlighted a systematic contextual effect across library members, likely a result of misfolding of the switch sequence with the coding sequence of mCherry. This was initially evident from high basal expression levels and low dynamic ranges across all library members (Supplementary Table 3) and individually characterized hits (Supplementary Fig. 12). Secondary structure prediction using RNAstructure[50] suggested that the coding sequence of the mCherry fluorescent reporter misfolded with part of the ribozyme stem and folinic acid aptamer (Supplementary Note 3). Thus, RNA-Seq was repeated with an eGFP reporter expression plasmid, which did not misfold with the switch library, and generated switches exhibiting low basal expression levels and dynamic ranges up to 4.9-fold (Supplementary Table 3). Six library members were selected for individual validation via flow cytometry using an eGFP reporter construct, and the validated folinic acid switch candidates exhibit activation ratios spanning 2.8–5.3-fold in relative fluorescence levels (eGFP/mCherry ratio) and basal expression levels as low as 12% (Fig. 4k).

RNA-Seq of cyclic di-GMP-II library yielded sequences with low basal expression levels, with moderate dynamic ranges of up to twofold (Supplementary Table 3). The cyclic di-GMP-I library exhibited lower activation ratio than cyclic di-GMP-II, due to higher basal RNA expression levels (39% compared with 17% for cyclic di-GMP-II; Supplementary Fig. 13; Supplementary Table 3). Four cyclic di-GMP-II library members were selected for individual validation via flow cytometry and the validated cyclic di-GMP-II switches exhibit activation ratios spanning 1.6–2.0-fold in relative fluorescence levels and basal expression levels <11% (Fig. 4l). Unlike the other aptamer–ligand pairs, the cyclic di-GMP ribozyme switches did not exhibit expression levels approaching full induction (i.e., levels matching sTRSVctl) even at high concentrations of the cognate ligand, possibly due to limited intracellular ligand availability. A similar observation was made with folinic acid switches, where induction levels were improved when the SLC46A1 transporter was overexpressed (Supplementary Fig. 9).

FACS-Seq assays were performed for each library to confirm that RNA-Seq accurately identified optimal gene-regulatory switches of protein expression (Fig. 4g–i; Supplementary Table 4; Supplementary Fig. 15). Preliminary analysis of the RNA-Seq data indicates that the N5 and N6 libraries exhibit similar numbers of sequences with low basal expression of reporter genes (Supplementary Fig. 14). Hence, the N5 libraries for FACS-Seq are maintained at full coverage (>100-fold) and to reduce library diversity, only the cells with the lowest 10% in basal expression from the N6 libraries were included at >100-fold. FACS-Seq and RNA-Seq results agree on 78%, 42%, and 79% of identified library members belonging to the top 10th percentile of activation ratios for xanthine, cyclic di-GMP-II, and folinic acid switches, respectively (Fig. 4g–i inset; Supplementary Fig. 6). The agreement between FACS-Seq and RNA-Seq for the cyclic di-GMP library is lower due to the lower fold change exhibited, such that the measurements are more influenced by noise. The FACS-Seq results for the new switch libraries lend further support for RNA-Seq data being predictive of the ribozyme switch's ability to regulate protein expression.

**Sequence and structural motifs of ribozyme switches.** Purine-rich sequence motifs define highly functional switches with low basal expression, and are conserved in an aptamer-dependent manner. We examined loop II sequences with five bases from theophylline, xanthine, folinic acid, cyclic di-GMP-II, and cyclic di-GMP I libraries, and determined the sequence motif (sequence logos by Shannon entropy) from the top five

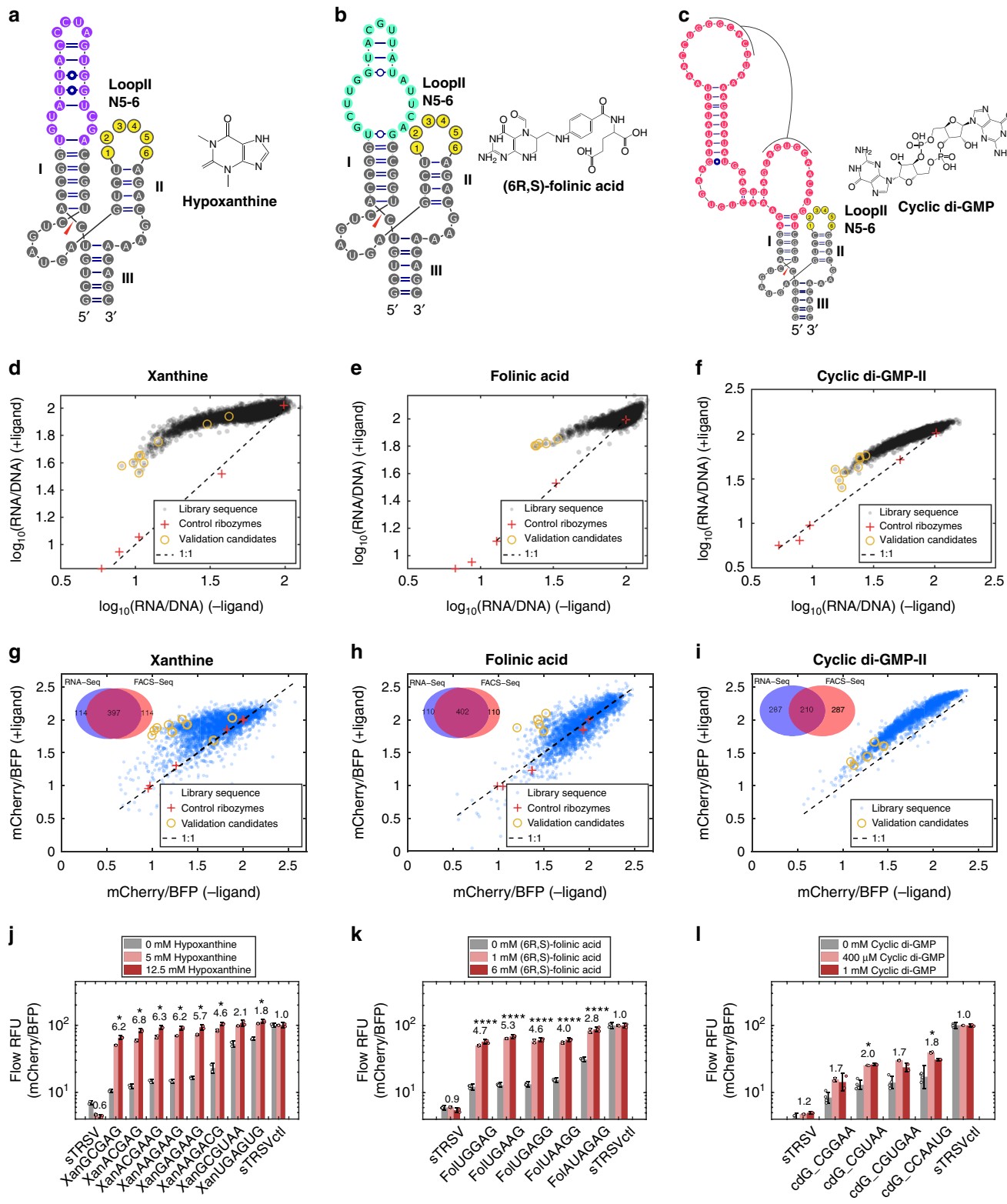

sequences with the lowest basal mRNA levels from the RNA-Seq results (Fig. 5a). Position 5 is highly conserved for either A or G, but not both, for each library. Position 1 is conserved whereby each library specifically prefers one of the four bases, possibly due to aptamer-loop interaction. A consensus motif of NʀRAR is found across the five libraries, where N represents a degenerate base of similar probabilities of A, C, U and G, R

represents purines (A or G), and ʀ at position 2 is only weakly conserved. A common motif suggests potential stabilizing effects within the loop, independent of any interactions with the aptamer. A boosting tree regression performed on the pooled libraries further quantifies the contribution of each sequence feature to the mRNA level (Fig. 5b) using SHAP (SHapley Additive exPlanation) values[51]. In agreement with the NʀRAR

**Fig. 4** RNA-Seq and FACS-Seq of xanthine, folinic acid and c-di-GMP switch libraries. **a–c** Secondary structures for ribozyme switch libraries designed with the **a** xanthine aptamer (purple), **b** folinic acid aptamer (green), and **c** the binding domain of the cyclic di-GMP-II riboswitch (red). All binding domains are grafted onto stem I of the sTRSV hammerhead ribozyme (gray), while stem loop II consists of 5–6 degenerate bases (yellow). **d**, **f** Normalized RNA read count from RNA-Seq assays performed on xanthine, folinic acid, and cyclic di-GMP-II ribozyme switch libraries in the absence and presence of ligand. −/+ Ligand conditions are **d** 0/5 mM hypoxanthine, **e** 0/6 mM (6R, S)-folinic acid, **f** 0/1 mM cyclic di-GMP. Pluses are spiked-in control ribozymes; circles indicate sequences selected for validation. **g–i** Relative fluorescence values from FACS-Seq assays performed on xanthine, folinic acid, and cyclic di-GMP-II ribozyme switch libraries in the absence and presence of ligand. −/+ Ligand conditions are **g** 0/5 mM hypoxanthine, **h** 0/6 mM (6R, S)-folinic acid, **i** 0/1 mM cyclic di-GMP. Inset Venn diagrams show the number of sequences shared between RNA-Seq and FACS-Seq for sequences with >90th percentile activation ratio. **j–l** Relative fluorescence values from flow cytometry analysis of individual sequences of the xanthine (**j**), folinic acid (**k**), and cyclic di-GMP-II (**l**) ribozyme switch libraries. Activation ratio of mCherry/BFP (xanthine and cyclic di-GMP) or eGFP/mCherry (folinic acid) at the higher ligand concentration is indicated above each set of bars. sTRSV is the wild-type hammerhead ribozyme; sTRSVctl is a non-cleaving mutant of sTRSV; error bars indicate standard deviation of two or more biological replicates; asterisks indicate the Benjamini–Hochberg corrected p-values of switching significance between 0 mM and the higher ligand concentration using p-values from unpaired, one-tailed t-tests, *p < 0.01, **p < 0.001, ***p < 0.0001, ****p < 0.00001. Filled circles are individual replicate data points. Source data are available in the Source Data file

motif, loop II-G5 provides the greatest impact (by SHAP value) in lowering the mRNA basal expression, followed by loop II-A5 and loop II-A4, whereas C at every position is detrimental for most sequences.

Exceptions to the NRRAR motif exist, likely due to base–base interactions. For instance, switches with highest activation ratio have U at position 3 (theophylline and cyclic di-GMP-II) and C at position 2 (xanthine). From analysis of pairwise base contributions to ribozyme activity (Supplementary Fig. 16), C1-U3 shows the lowest (−2.9) and second-lowest (−2.5) 5th percentile normalized mRNA levels for position 1–position 3 of theophylline and cyclic di-GMP-II libraries, respectively, even though U3 alone shows no contribution (position 3 major grid). Similarly, for xanthine, while A2 shows a greater overall effect of lowering the 5th percentile mRNA levels than C2, C2-A4 and C2-A5 lower mRNA levels to greater extents than A2-A4 and A2-A5. Such favorable pairwise interactions are likely to be aptamer dependent, implying possible triple base interaction involving specific bases in the aptamer, as these specific pairwise contributions are averaged out and absent in the pooled library analysis (Fig. 5c). In the pooled libraries, pairwise contributions common to all five libraries exist, e.g., G at position 1 is only favorable in lowering the 5th percentile mRNA level if A3, A4, or G4 is present; otherwise, A1 is the overall most favorable base at position 1.

To investigate loop II sequence features in the pooled library that contribute to different levels of ribozyme self-cleavage, we divided the library into 14 bins of normalized RNA levels, and performed Shannon entropy and mutual information analyses to identify sequence and structural motifs, respectively (Fig. 5d; Supplementary Fig. 17). A/G at position 5 is first enriched in sequences with low 5th percentile normalized RNA levels, whereas no other position shows any conservation. However, even with little conservation in Shannon entropy, mutual information for position 1 with positions 2, 3, and 4 are evident in the lowest 1st, 3rd, 4th, and 5th bins, while the 3rd and 4th positions have greater mutual information in the 2nd lowest bin. The analysis suggests that pairwise interactions may contribute to the structural stabilization of loop II tertiary interactions, either amongst the bases within loop II or across to the aptamer on loop I, or both.

The NRRAR motif serves as a design heuristic for semi-rational design of ribozyme switches with low basal expression and high activation ratios. Since positions 1 and 5 are highly conserved and specific to the aptamer sequence, one can first screen eight designs for each aptamer library design consisting of all four bases at position 1 and A/G at position 5, namely ARRAA, CRRAA, URRAA, GRRAA, ARRAG, CRRAG, URRAG, and GRRAG. Drawing from the RNA-Seq data for

the theophylline, xanthine, folinic acid, cyclic di-GMP-II, and cyclic di-GMP I libraries with 5 bases in loop II, the most optimal N1 and R5 pairs are shown to lower library mean mRNA level by 4.0, 3.7, 3.9, 3.3, and 2.3-fold, respectively, compared with a degenerate library of NNNNN (Fig. 5e). The remaining four sequence variants can be further screened, to achieve minimum basal levels of 11.8%, 10.4%, 10.6%, 17.3%, and 36.3%, corresponding to activation ratios of 5.9, 4.0, 4.9, 2.1, and 1.9, respectively. For an idealized ligand with full intracellular availability and perfect inhibition of ribozyme self-cleavage, the theoretical activation ratios would be 8.5, 9.6, 9.4, 5.8, and 2.8. Since the five aptamers tested here are highly diverse in sequence, size, structure, and origin (artificial evolution or natural riboswitches), this design heuristic could be applied to other aptamers that meet the binding and self-cleavage criteria of aptamer-ribozyme integration.

## Discussion
Highly functional ribozyme switches were generated via screening designed libraries that graft aptamer sequences onto stem I of the sTRSV hammerhead ribozyme and randomizing the opposing loop, with slight variations to the overlapping base pairs between aptamer and ribozyme stem. Library designs were selected by generating multiple library designs and then screening for retained (i) binding affinity and (ii) cleavage activity. In selecting between libraries that meet these criteria, the design exhibiting greater in vitro bulk cleavage should be favored to identify switches that exhibit higher switch activation ratios. As verified across RNA-Seq, FACS-Seq, and flow cytometry validation results, switch sequences with lower basal gene expression exhibit higher activation ratios.

Engineering of ribozyme switches requires consideration of ligand availability and aptamer interaction with the surrounding genetic context. For example, folinic acid ribozyme switches exhibit high basal mCherry RNA and protein expression levels, likely due to the folinic acid aptamer sequence interacting with the mCherry sequence. Replacing the mCherry reporter with an eGFP reporter lowered the basal expression sufficiently to improve the activation ratio of switches in the library. Either in silico screens for misfolding or in vitro cleavage assays performed with the entire transcript could be used to evaluate reporter compatibility before performing in vivo screens. Achieving full induction of the folinic acid switches required overexpression of the human SLC46A1 transporter in HEK293T to increase intracellular concentrations of folinic acid. Similarly, the cyclic di-GMP switches do not achieve full induction but there is no known transporter. Future studies can be performed to elucidate the transport mechanism or other

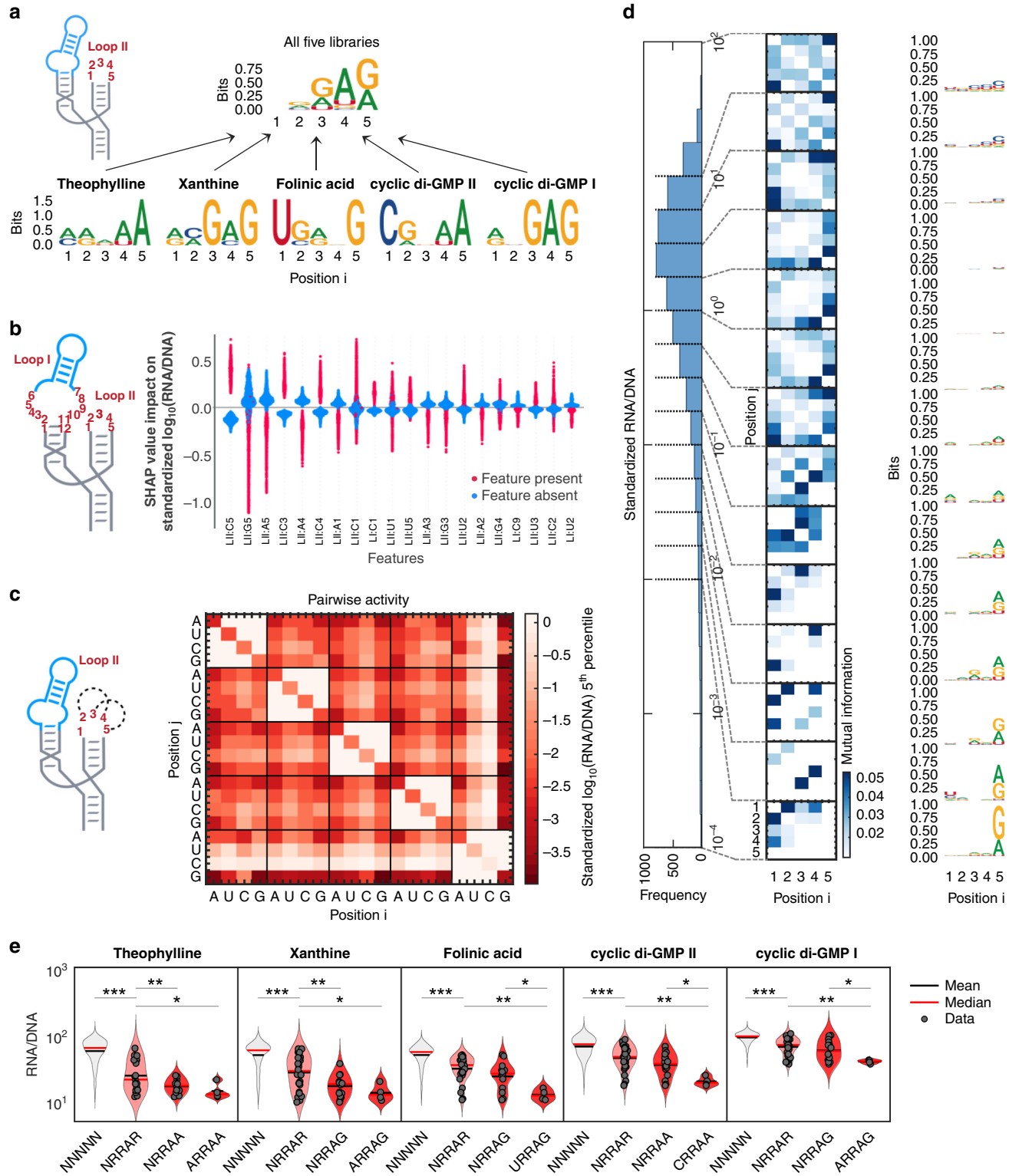

metabolic effects on this metabolite, which can be aided by our newly engineered cyclic di-GMP biosensor.

We developed a quantitative RNA-Seq approach to assay the gene-regulatory activity of ribozyme switch libraries in mammalian cells. To precisely measure variability in RNA levels resulting from switch activity, we identified DNA abundance variability within a library of synthetic oligonucleotides as the main source of dispersion, and showed that normalization of

RNA read count by DNA read count provides significant noise reduction. Additional sources of bias could result from insufficiencies in library coverage resulting in random sampling noise, such as low sequencing depth, *E. coli* plasmid transformation rate, or transfection rates. These biases can be minimized by scaling experiments to ensure at least 100–1000-fold library coverage at every experimental stage.

**Fig. 5** Sequence-function relationships and design principles. **a** Sequence logos representing Shannon entropy and consensus for five sequences with the lowest RNA/DNA levels from each loop II N5 library and for the combination of all five libraries. Schematic illustrates the position numbering on loop II. **b** SHAP (SHapley Additive exPlanation) values indicating the impact of the 20 most important sequence features (identity and position) on the predicted standardized $\log_{10}$(RNA/DNA) value of every sequence (dot) from five pooled libraries ($n = 5093$ N5 library variants), derived from a boosting tree regression model (XGBoost). Sequence features are derived from loop I and loop II as indicated with position numbering in the adjacent schematic. **c** Pairwise contribution to the 5th percentile standardized $\log_{10}$(RNA/DNA) ratio of the five pooled libraries for each of A, U, C, G nucleotides at two positions within loop II. Schematic indicates position numbering of the feature analyzed, with dashed lines as examples of pairwise interactions. **d** The five pooled libraries divided into 14 activity bins according to different levels of standardized $\log_{10}$(RNA/DNA). For sequences in each bin, mutual information analysis is indicated for a pair of nucleotides in loop II in the heat map, and the sequence logo of the sequences in the same bin is shown to the right. **e** Distribution of normalized mRNA levels RNA/DNA of different ligand libraries when constrained by conserved loop II N5 motifs indicated in the x-axis. N indicates the presence of all bases A, U, C, G; R indicates the presence of only A and G; asterisks indicate significance from one-tailed t-tests, $*p < 0.05$, $**p < 0.005$, $***p < 10^{-9}$. Filled circles are individual sequence data points

Recent work using RNA-Seq to characterize the ribozyme cleavage activity of thousands of sequences in mammalian cells expressed pistol ribozymes directly from a U6 promoter to achieve high RNA yields[33]. Total RNA recovered from transfected cells were gel extracted to isolate cleaved and uncleaved fragments of a known size prior to NGS analysis to determine cleavage ratio. Our RNA-seq method directly measures the impact of ribozyme activity on RNA polymerase II-transcribed intact mRNA levels and subsequent protein expression levels, which is a more relevant context for gene expression regulation, without requiring gel-based separation of RNA molecules. We include a benchmark set of control ribozymes in each library, regardless of aptamer length (ranging from 20 nts for xanthine to 80 nts for cyclic di-GMP I) to standardize the measurement across different libraries and experiments.

We performed FACS-Seq to verify that RNA-Seq results can identify top performing switches at the protein regulatory-level. However, the FACS-Seq results exhibit greater noise compared with RNA-Seq. Noise may be introduced through high rates of lentiviral recombination[52,53], epigenetic silencing[54], and semi-random integration at genomic sites[55,56]. Anti-repressor insulating sequences can be used to mitigate this undesired epigenetic silencing[57], and greater library coverage can be maintained during library propagation, albeit at the cost of greater time and labor. Since the mammalian cell RNA-Seq assay can be performed with ease and accuracy within a much shorter time period, it offers a better workflow for rapid design and testing of highly functional ribozyme switches.

RNA-Seq data provide insight into key sequence-function relationships underlying ribozyme switch regulatory activities. Conserved sequence motifs were observed across all libraries in positions 3, 4, and 5, while positions 1 and 5 are aptamer dependent. Base–base interactions within loop II contribute to different regulatory activities. While we derived a design heuristic of using the NRRAR sequence motif for rapid identification of near optimal switch sequences, fully rational design strategies may be possible in the future as expanded aptamer sequence space and base–base interaction sequence-function landscape are explored.

We developed a massively parallel RNA-Seq approach to simultaneously assay mRNA and protein expression regulatory activities of tens of thousands of library sequences to identify highly functional ribozyme switches in mammalian cells. Ribozyme switches generated through this approach can be used directly in different systems or coupled with downstream signal processing elements[7,45,58] to achieve application-specific performance requirements. Although the RNA-Seq assays are limited by cell-based transfection efficiencies, this throughput will facilitate future engineering efforts to further improve the activities of ribozyme switches as gene control elements. For example, the assays can be extended to library designs investigating newer classes of ribozymes such as twister, twister sister and pistol ribozymes, which may

exhibit faster cleavage rates than hammerhead ribozymes. The quantitative RNA-Seq approach provides faster and more accurate measurements of library performance, accelerating the design-build-test cycle of engineering different ribozyme switch devices, while generating massive datasets to further our understanding of the sequence-function relationship toward rational design. This approach will significantly expand the repertoire of devices for applications in controllable gene and cellular therapy, perturbing molecular pathways in studying cell biology, and construction of multilayer gene network and circuits for engineering complex cellular computation.

## Methods

**DNA oligonucleotides, plasmids, and reagents.** DNA synthesis of templates for in vitro co-transcriptional cleavage assays and binding affinity assays, library sequence templates, and Illumina sequencing preparation primers were obtained from Integrated DNA Technologies (Coralville, IA, USA). All other cloning related primer sequences were obtained from Protein and Nucleic Acid Facility (Stanford, CA, USA). All plasmids were constructed using standard molecular biology cloning techniques. Unless otherwise specified, enzymes for cloning, including restriction enzymes and Gibson assembly components, were obtained from New England Biolabs (Ipswich, MA, USA). Unless otherwise specified, all PCR reactions were performed using the Kapa HiFi Hotstart Kit from Roche (Basel, Switzerland) following the manufacturer's instructions. All cloned constructs were sequence verified by Elim Biopharmaceuticals (Hayward, CA, USA) or QuintaraBio (Albany, CA, USA). Plasmids were purified using Wizard® Plus SV Minipreps DNA Purification Systems from Promega (Madison, WI, USA) following the manufacturer's instructions.

**Ligand preparations.** Theophylline (catalog number T1633) and hypoxanthine (catalog number H9377) were obtained from Sigma-Aldrich (St Louis, MO, USA). Cyclic diguanylate monophosphate was obtained as cyclic-di-GMP sodium salt from Sigma-Aldrich (catalog number SML1228-1UMO) or Fisher Scientific (Hampton, NH, USA; catalog number NC0432586). (6R, S)-folinic acid (catalog number 16.22) was obtained from Schircks Laboratories (Jona, Switzerland). Theophylline was dissolved in water to make a 25 mM concentrated stock solution. A concentrated hypoxanthine stock solution was made by dissolving 1 M hypoxanthine into 1 M NaOH and then diluting to a 200 mM concentrated solution in water. Folinic acid was directly dissolved in reaction buffer or growth media to make a 6 mM stock solution. Cyclic di-GMP was directly dissolved in reaction buffer or growth media to make a 1 mM stock solution. All stock solutions were diluted to indicated concentrations in appropriate solutions. All ligand solutions were filter sterilized before use. All ligand solutions were freshly prepared before each experiment or stored at 4 °C for less than a week.

**Cell culture.** HEK293T cells from ATCC (Manassas, VA, USA) were used in all experiments. Cells were cultured in Dulbecco's modified Eagle's medium (DMEM) (GlutaMax), 100 U/ml penicillin, 100 mg/ml streptomycin, and 10% fetal bovine serum, and maintained at a minimum cell density of 0.11 million/ml in 10 cm dishes. All cell culture reagents were obtained from Thermo Fisher Scientific unless otherwise specified.

**Ribozyme switch library design and construction.** In coupling the aptamer to the hammerhead ribozyme, the base stem of each aptamer was truncated to 0, 1, or 2 base pairs to join with 6, 5, or 4 base pairs, respectively, from stem I of the sTRSV hammerhead ribozyme. Some designs truncated the base stem of the aptamer to 0 or 1 base pairs to join with 4 or 3 base pairs, respectively, from stem II of the sTRSV hammerhead ribozyme. The library designs assume the following

architecture,

$$5' - GCTGTCACCG\Delta\Delta <aptamer\ or\ N4-6>$$
$$\Delta\Delta CGGTCTGATGAGTCA < N4 - 6\ or\ aptamer > \Delta GACGAAACAGC - 3'.$$

where $\Delta$ represents a nucleobase being optimized, and N4–6 are degenerate loop sequences of length 4–6 nucleotides. For each design, secondary structure folding was performed using a stochastic folding algorithm in RNAstructure[50] to verify that the integrated aptamer-ribozyme sequence preserves the active conformation of the ribozyme domain and the structure of the original aptamer domain. Sequences for library designs, including aptamer domains and predicted secondary structures, are shown in Supplementary Fig. 7.

Library oligonucleotides were synthesized by Integrated DNA Technologies. Library oligonucleotides were ordered as two pieces, 5′-AAACAAACAAAGCTGT CACCGΔΔ <aptamer> ΔΔCGGTCTGATGAGTCΔ-3′ and 5′-TTTTTATTTTTCT TTTTGCTGTTTCGTCΔ NNNNN ΔGACTCATCAGACCGΔ-3′, or 5′-AAACA AACAAAGCTGTCACCGΔΔ NNNNN ΔΔCGGTCTGATGAGTCΔ-3′ and 5′-TT TTTATTTTTCTTTTTGCTGTTTCGTCΔ <aptamer> ΔGACTCATCAGACC GΔΔ-3′. The randomized regions were synthesized with hand-mixing for equal representation of degenerate bases. To make the double stranded library shown in Supplementary Table 1, library sequences were PCR amplified with Kapa HiFi HotStart PCR Kit (Roche), with each of the two pieces at 500 nM, and using the GC buffer supplemented with 1 M betaine monohydrate. PCR was performed for three cycles, denaturing at 98 °C, annealing at 55 °C, and elongating at 72 °C according to manufacturer's instructions, and successful reactions were verified on a 3% agarose gel via electrophoresis.

**Surface plasmon resonance assay**. To prepare library sequences for SPR binding assays, the previously prepared double stranded PCR product was diluted 1:25 as template into another PCR reaction using the Kapa HiFi HotStart PCR Kit (Roche) and 400 nM each of primers 5′-AATTTAATACGACTCACTATAG GGAAACA AACAAAGCTGTCACCG-3′ and 5′-TTTTTTTTTTTTTTTTTTTTTTTTGCTG TT TTGTC-3′, to append the T7 promoter and the poly-A sequence for hybridizing transcribed RNA molecules to the polyT sequence on the sensor chip. The second primer also incorporates a G12A mutation into the catalytic core of the ribozyme to prevent the transcribed library RNA from cleaving. PCR was performed for 10 cycles, denaturing at 98 °C, annealing at 55 °C, and elongating at 72 °C according to manufacturer's instructions. The PCR product was purified using the DNA Clean & Concentrator kit (Zymo Research) and the concentration was quantified using NanoDrop (Thermo Fisher Scientific). Transcription was performed using the MEGAshortscript™ T7 Transcription Kit (Thermo Fisher Scientific) according to manufacturer's instructions, using 100 nM DNA template and incubating at 37 °C for 5 h. Transcribed RNA was purified using the RNA Clean & Concentrator Kit (Zymo Research) according to manufacturer's instructions and quantified on a NanoDrop Spectrophotometer (Fisher Scientific).

Binding affinity characterizations of transcribed ribozyme switch libraries were performed using a SPR assay on the Biacore X100 from GE Healthcare (Chicago, IL, USA) instrument as previously described[49]. Briefly, a Biacore CM5 sensor chip (GE Healthcare) was immobilized with a polyT sequence 5′-/5AmMC6/TTTTTT TTTTTTTTTTTTTT TTTTTTTTTTTTTTTTTT-3 (Integrated DNA Technologies), using the amine coupling kit (GE Healthcare) according to the manufacturer's instructions. The transcribed RNA was diluted into the running buffer for the SPR assay (HBS-N buffer (0.1 M HEPES, 1.5 M NaCl, pH 7.4, GE Healthcare) and 0.5 mM MgCl₂ (Thermo Fisher Scientific)), to provide ~2.5 μg RNA per cycle in the run protocol. Ligands were dissolved directly into the running buffer, with the exception of hypoxanthine, which was first dissolved at 1 M in 1 M NaOH, before being diluted into the running buffer. The multicycle kinetics protocol was used for the assay which consists of (1) capturing the RNA onto the sensor chip with 40 s contact time at a 5 μl/min flow rate, (2) associating and dissociating the ligand for 120 s each at a 30 μl/min flow rate, and (3) regenerating the sensor chip using 25 mM NaOH with a 30 μl/min flow rate for 30 s. The SPR sensorgrams for cycles with ligands were background subtracted by the blank cycle, where running buffer without ligand was flowed over the captured RNA. The background subtracted sensorgram results were evaluated with the Biacore X100 Evaluation Software, and equilibrium dissociation constants were determined from the model, $R = [L]*R_{max}/([L] + K_D)+$ offset, where R is the SPR response unit, $R_{max}$ is the peak SPR response unit, $[L]$ is ligand concentration, and $K_D$ is the dissociation constant.

**Gel electrophoresis characterization of cleavage activities**. In vitro co-transcriptional cleavage reactions were performed on libraries to obtain an initial assessment of cleavage activity. DNA templates encoding ribozyme switch libraries (Supplementary Table 1) were PCR amplified using Kapa HiFi Hotstart Kit (Roche) according to manufacturer's instructions, using primers 5′-AATTTAAT ACGACTCACTATAGGGAAAC AAACAAAGCTGTCACCG-3′ and 5′-TTTTTT TTTTTTTTTTTTTTTTTTGCTGTTTCGTC-3′. The resulting PCR products were purified using the DNA Clean & Concentrator Kit (Zymo Research) according to manufacturer's instructions and subsequently quantified using NanoDrop (Thermo Fisher Scientific). Transcription reactions were performed with T7 RNA polymerase and accompanying buffer (New England Biolabs), 9 mM each of ribonucleotide mix (New England Biolabs), 10 mM Dithiothreitol from Gold

Biotechnology (St. Louis, MO, USA), 1 U Superase•IN (Life Technologies), and 40 nM of the purified DNA template. Transcription reactions were performed for 15 min by incubating at 37 °C in the presence and absence of the ligand (Supplementary Fig. 2). Reactions were stopped by adding four volumes of Tris•EDTA solution (Integrated DNA Technologies).

The RNA products from the co-transcriptional cleavage reactions were analyzed by PAGE. 2x TBE-urea sample loading buffer from BIO-RAD (Hercules, CA, USA) was added to each reaction tube. Samples were denatured by incubating at 95 °C for 10 min. Five microliters of each sample was loaded onto a denaturing 10% polyacrylamide gel with 8 M Urea in a Mini-PROTEAN® Electrophoresis System (BIO-RAD). Electrophoresis was performed at 240 V for 45 min. The gel was subsequently stained with the GelRed loading dye from Biotium (Fremont, CA, USA) for 15 min before imaging with the Ethidium Bromide setting directly on a GeneSys G:Box fluorescent gel imager from Synoptics (Frederick, MD, USA). The FIJI implementation of ImageJ, version 2.0.0-rc-68/1.52e, was used to quantify the intensities $I$ of the gel bands. The fraction cleaved (Supplementary Fig. 8) was determined by:

$$\text{fraction cleaved} = \frac{(I_{uncleaved} - I_{background})/\text{length}_{cleaved}}{(I_{cleaved} - I_{background})/\text{length}_{cleaved} + (I_{uncleaved} - I_{background})/\text{length}_{uncleaved}} \quad (1)$$

The fold change in fraction cleaved and fraction uncleaved were determined by:

$$\Delta \text{ fraction cleaved} = \frac{\text{fraction cleaved}_{with\ ligand}}{\text{fraction cleaved}_{without\ ligand}} \quad (2)$$

$$\Delta \text{ fraction uncleaved} = \frac{\text{fraction uncleaved}_{with\ ligand}}{\text{fraction uncleaved}_{without\ ligand}} \quad (3)$$

where fraction uncleaved was determined by:

$$\text{fraction cleaved} = \frac{(I_{uncleaved} - I_{background})/\text{length}_{uncleaved}}{(I_{cleaved} - I_{background})/\text{length}_{cleaved} + (I_{uncleaved} - I_{background})/\text{length}_{uncleaved}} \quad (4)$$

**Library plasmid cloning for intracellular expression**. A *mCherry-BFP* dual expression cassette pCS4076 (Supplementary Fig. 18) was modified from pCS2587 (gift from Melina Mathur, Christina Smolke laboratory), which has EF1α_promoter-BFP_codingsequence-HSVTK_polyA cloned into BglII and CMV_promoter-mCherry_codingsequence-BGHpolyA cloned into BamHI/NotI of pcDNA5/FRT (Thermo Fisher Scientific), to include restriction sites BamHI and KpnI for cloning individual and library sequences. This expression vector was used for all transient transfection assays for validation of individual sequences and RNA-Seq experiments. For some folinic acid experiments (Supplementary Note 3), the *eGFP* expression plasmid vector pCS408 was used in RNA-Seq and flow cytometry validation experiments.

We designed a lentiviral transfer plasmid (pCS4077) which was suitable for screening ribozyme switches. The expression cassette was assembled to contain the CMV promoter, *mCherry* coding sequence, ribozyme library, and BGH poly-A tail sequence in an antisense orientation to the lentiviral 5′ LTR promoter-controlled transcript. This design avoids degradation of transcripts harboring faster-cleaving ribozymes, which would lead to underrepresentation of those elements in the final library of transduced cells. The same lentiviral construct also contains (in the sense orientation relative to the 5′ LTR transcript) an EF1α promoter-controlled BFP reporter, which serves as a transduction marker and as a control to normalize for general gene expression noise of *mCherry*. To construct the lentiviral destination vector, the *mCherry-BFP* cassette from pCS4076 was PCR amplified using the Expand™ High Fidelity PCR System (Roche) according to manufacturer's instructions. The lentiviral vector backbone was obtained from pKL5 (gift from Keara Lane, Markus Covert laboratory), which was digested with restriction enzymes AgeI and XbaI. The PCR product of the *mCherry-BFP* cassette was assembled into the cut pKL5 vector using a standard Gibson assembly protocol[59], in antisense direction to the lentiviral 5′ LTR promoter-driven transcript of the lentiviral expression cassette, to prevent ribozyme self-cleavage in the viral packaging process. The *BFP* expression cassette was excised from this vector with restriction enzymes NsiI and XmaI (New England Biolabs) and reinserted in the reverse orientation via Gibson cloning to make the final vector pCS4077 (Supplementary Fig. 18).

Library sequences (Supplementary Table 1) and control ribozymes were PCR amplified with Kapa HiFi HotStart PCR Kit (Roche) to add on flanking regions for Gibson assembly. For cloning into pCS4076, the primers used were pCS4076-N6_ACCG_FWD and pCS4076-N6_GTC_REV. For cloning into pCS4077, the primers used were pCS4077-N4_ACCG_FWD and pCS4077-N4_GTC_REV. For cloning into pCS408, the primers used were pCS408-ACCG_FWD and pCS408-GTC_REV. The PCR products were purified with the DNA Clean & Concentrator kit (Zymo Research) and their concentrations were determined with NanoDrop (Thermo Fisher Scientific). Each of the control ribozymes was mixed in at twice the representation per sequence as the individual library members according to molar ratios.

The vector pCS4077 was digested with BamHI-HF and XbaI (New England Biolabs), pCS4076 was digested with BamHI-HF and KpnI-HF (New England Biolabs), and pCS408 was digested with XbaI and XhoI. The digested vectors were gel extracted and purified with the DNA Clean & Concentrator™-5 kit (Zymo

Research) according to manufacturer's instructions. The purified PCR library product with spiked-in control ribozymes was then Gibson assembled into the digested vector. For each library, 500 pM of purified cut vector and 12.5 nM of insert was added to a 40 μl Gibson reaction. Gibson reactions were incubated at 50 °C for 4–16 h. The Gibson assembled product was column purified and concentrated to at least 100 ng/μl. One hundred nanograms of the purified Gibson product was transformed into 20 μl ElectroMAX Stbl4 Competent Cells (Thermo Fisher Scientific) via electroporation according to manufacturer's instructions. Transformed cells were recovered in 1 ml SOC media provided by the ElectroMAX kit for 90 min with shaking at 30 °C. Cells were then inoculated into 500 mL LB liquid media with carbenicillin (100 μg/ml) at a 1:500 dilution. A dilution series was also performed and plated onto LB agar plates with carbenicillin (100 μg/ml) to determine transformation efficiency, where ~100–500X coverage is desired to ensure library full coverage. Library plasmids were prepared from *Escherichia coli* cultures with the Qiagen Plasmid Plus Midi Kit following the low-copy plasmid protocol.

**Targeted, quantitative RNA-Seq.** HEK293T cells were seeded in 6-well plates at 400,000 cells per well 18–24 h prior to transfection. The appropriate ligand was added 1 h prior to transfection either by dissolving the ligand directly in DMEM +10% FBS and sterilizing via filtration or by making a sterilized concentrated stock in water and subsequently diluting to the appropriate concentration in the media (Supplementary Table 3). Four micrograms of library plasmid was transfected per 6-well plate well using Lipofectamine 2000 transfection reagent (Thermo Fisher Scientific). Seventy-two hours post transfection, total RNA was extracted using the RNeasy Plus kit with QIAshredder (Qiagen) for homogenization following manufacturer's instructions. 10 mM EDTA (Thermo Fisher Scientific) was added to buffers RLT plus and RW1 to minimize ribozyme self-cleavage during the extraction process, with the final product eluted in 10 mM Tris-HCl (pH 8.0, Ambion) with 2 mM EDTA.

The reverse transcription primer (RT1 or RT2; Supplementary Table 5) was annealed with the purified total RNA in duplex buffer (50 mM NaCl, 10 mM Tris-HCl (pH 8.0) and 2 mM EDTA (Ambion)), with brief vortexing at room temperature. Reverse transcription was performed using the Omniscript Reverse Transcription kit (Qiagen) according to the manufacturer's instructions and incubated for 1 h at 37 °C using primer RT1 or RT2. cDNA yields were determined with the Kapa Library Quantification Kit (Roche). The RNA sample was also quantified with qPCR as a negative control to determine background DNA template concentration or any non-specific priming effect. The forward primer was mCherry_FWD or EGFP_FWD (Supplementary Table 5) with pCS4076 or pCS408, respectively, and the reverse primer was RT1 or RT2, depending on the primer used during reverse transcription. At least 20-fold RNA yield from the non-reverse transcribed samples was required before further downstream processing. If the minimum yields were not met, the transfection efficiency was re-optimized to ensure >20% transfected cells. The cDNA was cleaned up and concentrated using the DNA Clean & Concentrator kit (Zymo Research) according to the manufacturer's instructions (Zymo Research).

With the same primers for qPCR quantification, an outer PCR was performed for 24 cycles, using the Kapa HiFi Hotstart Kit with GC buffer (Roche), with annealing at 55 °C. Primers with Illumina adapter sequences and custom barcodes (Supplementary Table 5) were used to barcode the sequences using the same PCR settings, for seven cycles of amplification. With a 1:25 dilution of the barcoding PCR product, a final PCR was performed using short Illumina i5 and i7 adapter sequences for another eight cycles to ensure the library was full length. The library was size-selected and purified using 2% agarose gel cassettes in the Pippin prep system (Sage Science) and quantified using the dsDNA HS Assay Kits on the Qubit 2.0 fluorometer according to manufacturer's instructions. The sample was size verified on the Agilent 2100 Bioanalyzer using the High Sensitivity DNA assay according to manufacturer's instructions. This sample was mixed with 10% PhiX before sequencing using the MiSeq v3 2 × 75 reagent kit on an Illumina MiSeq sequencer at the Stanford Protein and Nucleic Acid facility (PAN). When a greater number of reads were desired, sequencing using the 2 × 75 NextSeq mid-output or high-output reagent kit on the NextSeq sequencer was performed at the CZ Biohub, similarly with 10% PhiX mixed in with the library sample.

**Quantitative FACS-Seq.** Lentiviral library plasmids were transfected into a HEK293T cell line with the packaging (pMO86) and envelope (pMO87) plasmids using the calcium phosphate transfection protocol[60] to make lentiviral particles, and media was replaced 24 h later. Viruses were harvested 2 days later by collecting the supernatant. HEK293T cells to be transduced were seeded at a cell density of 1.1 million per 10 cm dish. Eighteen to twenty-four hours later, cells were transduced with freshly harvested virus with 8 μg/ml polybrene obtained from Santa Cruz Biotech (Dallas, TX, USA). About 1.74 million cells in each of two 10 cm dishes were transduced with 6 ml of unconcentrated virus to ensure the number of cells transduced (~10–20% of the population transduced) have a more than 200-fold coverage of the theophylline library with 1280 diversity. Only 10–20% of cells were transduced to optimize for single integration events. This 10 cm dish unit was scaled in discrete steps for libraries with greater diversity (Supplementary Table 6). During any passaging of the transduced cells, cells were plated at a cell density of

~0.35 million/ml, i.e., with at least the same number of cells used for transduction to maintain library coverage.

All fluorescence activated cell sorting (FACS) experiments were performed on the BD Influx cell sorter at the Stanford FACS facility (Supplementary Fig. 19). BFP is excited by the 405 nm violet laser, and uses the 460/50 BP filter. *mCherry* is excited by the 561 nm yellow laser, and uses the 610/20 BP filter. Cells infected with the virus were grown for 3–5 days before sorting to enrich for transduced cells. Following the first enrichment, cells were grown for 4–6 days before the transduced cells were enriched for the 5–10% fastest cleaving library members. Three to four days later, cells were incubated with the ligand of interest (Supplementary Table 4) for 48 h before FACS being sorted into 6 bins with evenly distributed activity levels. The bin widths were adjusted such that sTRSVctl evenly spans the two bins with the highest mCherry/BFP ratio; the bin widths also account for sTRSV spanning the two lowest bins. The binning gates were generated using a custom MATLAB script, available at https://github.com/jsxiang/Influx-custom-gates. Cells were sorted to give at least 200-fold coverage of estimated library diversity (Supplementary Table 7).

Cells from the binning sort were grown for another 5 days to expand 50–100-fold before DNA extraction using the DNeasy Blood & Tissue kit (Qiagen) according to manufacturer's instructions. The genomic DNA was restriction digested with SpeI and EcoRV (New England Biolabs) and the library construct were gel extracted to ensure that yields of DNA were at least sevenfold coverage of the number of sorted cells, as quantified by qPCR using the using the Kapa Library Quantification Kit (Roche). Outer PCR primers Outer_REV and mCherry_FWD were used at a final concentration of 400 nM. All qPCR reactions were performed on the BIO-RAD iCycler. PCR was performed to amplify the library sequences from the extracted DNA using the Kapa HiFi Hotstart Kit with GC buffer (Roche) and 0.5 M betaine monohydrate, annealing at 55 °C, for 35 cycles. The amplified products from each bin were PCR amplified with custom barcode primers using the same PCR settings for 6–8 cycles followed by quantification using the dsDNA HS assay kit on the Qubit® 2.0 Fluorometer according the manufacturer's instructions. Barcoded sequences were mixed according to the desired ratios, such that representation by each barcode correlates with the number of cells sorted in each bin. Mixed down PCR products were column purified using the DNA Clean & Concentrator kit (Zymo Research). A final PCR was performed for 6–8 cycles with Illumina i5 and i7 adaptor primers to ensure samples were of uniform length. The library was size-selected and purified using 2% agarose gel cassettes in the Pippin prep system (Sage Science) and quantified using the dsDNA HS assay kit on the Qubit® 2.0 Fluorometer according to manufacturer's instructions. The sample was size verified on the Agilent 2100 Bioanalyzer using the High Sensitivity DNA assay according to manufacturer's instructions. The prepared sample was mixed with 10% PhiX and sequenced on the Illumina Miseq sequencer using the MiSeq v3 2 × 75 reagent kit and run for 2 × 85 cycles.

**Sequencing analysis.** For the RNA-Seq analysis, paired-end sequence reads from FastQ files were first joined using PEAR version 0.9.6 using default parameters. Sequences that were too long but contain the known aptamer sequence were pasted together in a shell script. Sequences were filtered for the exact library architecture and spiked-in control ribozymes, before being demultiplexed using a custom Python script (available at https://github.com/jsxiang/processFASTQ) according to the experimental conditions, and the number of sequence reads per unique sequence for each experimental condition were counted. A custom MATLAB script was used for downstream analysis and is available at https://github.com/jsxiang/Ribozyme-RNA-seq. A threshold DNA read count of at least 100 sequencing reads per condition was set to select sequences for subsequent analysis. To determine the regulatory activity of each library sequence $i$, raw RNA read counts were first normalized to DNA read counts to determine the normalized RNA/DNA ratio then divided by that of the spiked-in sTRSVctl, and scaled by a factor of 100,

$$\mathrm{RD}_i = \frac{\left(\frac{\text{RNA read count}_i}{\text{DNA read count}_i}\right)}{\left(\frac{\text{RNA read count}_{\text{sTRSVctl}}}{\text{DNA read count}_{\text{sTRSVctl}}}\right)} \times 100 \qquad (5)$$

Unless otherwise indicated, all normalized RNA/DNA ratios are reported on a $\log_{10}$ scale. If sTRSVctl was not spiked-in, the mean of the 25 sequences with the highest RNA/DNA ratios were used for normalization for each library.

For FACS-Seq analysis, after paired-end sequences were joined with PEAR and filtered for sequences containing the library architecture, sequences were demultiplexed according to the barcodes using a custom Python script and the number of sequence reads per unique sequence for each activity bin was counted. A MATLAB script, available at https://github.com/jsxiang/Ribozyme-FACS-seq, was used to fit the bin counts to a normal distribution to obtain the fit parameter $\mu$ which corresponds to the mean relative fluorescence levels $\log_{10}$(mCherry/BFP). $\mu$ values were then fit to scale with fluorescence units as determined from flow cytometer analyzer using the five spiked-in control ribozymes, before normalizing to that of sTRSVctl and scaling by 100.

Statistical testing for significance in switching made use of unpaired, two-tailed $t$-tests implemented in MATLAB, and $p$-values for each set of with/without ligand comparisons were adjusted for multiple comparisons using the method of Benjamini and Hochberg using the MATLAB command mafdr ([list of $p$-values], 'BHFDR', 'true').

**Cloning of individual ribozymes and ribozyme switches**. Individual ribozymes and ribozymes switches were cloned into pCS4076 or pCS408 (Supplementary Table 8). All DNA templates for control ribozymes have the architecture 5′-AAACAAACAAAGCTGTCACCGGA <loop I> TCCGGTCTGATGAGTCC <loop II> GGACGAAACAGCAAAAAGAAAAATAAAAA-3′. Oligonucleotides for control ribozymes were synthesized by Integrated DNA Technologies as two pieces, 5′-AAACAAACAAAGCTGTCACCGGA <loop I> TCCGGTCTGATGA GTCC-3′ and 5′- TTTTTATTTTTCTTTTTGCTGTTTCGTCC <loopII> GGAC TCATCAGACCGGA-3′. Oligonucleotides for all ribozyme switches were synthesized by Integrated DNA Technologies as two pieces, 5′-AAACAAACAAA GCT GTCACCGΔΔ <aptamer> ΔΔCGGTCTGATGAGTC-3′, and 5′-TTTTTATTTT TCTTTTTGCTGTTTCGTCΔ ÑÑÑÑÑ(Ñ) ΔGACTCATCAGACCGΔΔ-3′, where Δ's are optimized bases for each library sequence, and Ñ's are fixed sequences. PCR amplification was performed using the Kapa HiFi Hotstart Kit with GC buffer (Roche) according to manufacturer's instructions, with each of the two pieces at 500 nM, and using the GC buffer supplemented with 1 M betaine monohydrate. PCR was performed for three cycles, denaturing at 98 °C, annealing at 55 °C, and elongating at 72 °C according to manufacturer's instructions.

A subsequent PCR was performed for 10–12 cycles with a 1:25 dilution of the unpurified product from the first PCR as template, and 400 nM of primers 5′-TA ACTGATCATAAATATAGGGCCCGGATCCAAACAAACAAAGCTGTCACC G-3′ and 5′-AGGCTGATCAGCGGGTTTGGTACCTTTTTATTTTTCTTTTTGC TGTTTCGTC-3′ using the same conditions as the first PCR (Supplementary Table 8). The PCR product was purified using the DNA Clean & Concentrator kit (Zymo Research) according to manufacturer's instructions and Gibson assembled into the plasmid vector pCS4076, which was digested with BamHI and KpnI (New England Biolabs). One Shot™ TOP10 Chemically Competent *E. coli* (Thermo Fisher Scientific) cells were transformed with the Gibson assembly product and plated on LB agar plates with carbenicillin (100 μg/ml) and incubated at 37 °C for 12–16 h. Individual colonies were sequence verified and inoculated in liquid LB media with carbenicillin (100 μg/ml). Plasmids were purified from cultures grown for 12–16 h using the Wizard® Plus SV Minipreps DNA Purification System kit (Promega) according to manufacturer's instructions. For cloning ribozyme sequences into the pCS408 plasmid vector, the second PCR was performed using primers 5′-GCAT GGACGAGCTGTACAAGTAACTCGAGAAACAAACAAAGCTGTCACCG-3′ and 5′-AGCGGGTTTAAACGGGCCCTCTAGATTTTTATTTTTCT TTTTGCT GTTTCGTC-3′, and the vector was digested with XhoI and XbaI.

**qPCR measurement of transcript levels**. HEK293T cells were seeded at 50,000 cells per well of a 24-well plate 24 h prior to transfection. Transfection of control ribozyme plasmids in the pCS4076 vector was performed using the Lipofectamine 2000 transfection reagent (Thermo Fisher Scientific) according to manufacturer's instructions, and 600 ng of plasmid were used in each transfection. Seventy-two hours later, total RNA was extracted using the RNeasy Plus kit with QIAshredder (Qiagen) for homogenization following manufacturer's instructions. 10 mM EDTA (Thermo Fisher Scientific) was added to buffers RLT plus and RW1 to minimize ribozyme self-cleavage during the extraction process, with the final product eluted in 10 mM Tris-HCl (pH 8.0, Ambion) with 2 mM EDTA.

Reverse transcription primer (RT1 and RT2) was annealed with the purified total RNA in 50 mM NaCl, 10 mM Tris-HCl (pH 8.0) and 2 mM EDTA (Ambion), with brief vortexing at room temperature. Reverse transcription was performed using the Omniscript Reverse Transcription kit (Qiagen) according to the manufacturer's instructions and incubated for 1 h at 37 °C. The Kapa Library Quantification Kit (Roche) was used to quantify the yields of the reverse transcription product, which was diluted 10-fold into the qPCR reaction. The forward primer used was mCherry_FWD and the reverse primer was the primer used during the reverse transcription reaction, RT1 or RT2. BFP transcripts were also reverse transcribed using the primer 5′-TCAATTAAGCTTGTGCCCC-3′ and qPCR to quantify the reverse transcribed cDNA used primers 5′-AGGCCTTCAC CGAGACCCTGTACCCCGCTGACGGCGGCCTGGAAGGCAGAAAAGGCCTT CACCGAGAC-3′ and 5′-TCAATTAAGCTTGTGCCCC-3′. To account for discrepancies in transfection efficiency from different wells, the concentrations of the *mCherry*-linked transcripts were normalized to the expression of BFP and scaled to set sTRSVctl as 100 as follows:

$$\text{normalized mCherry RNA concentration} = \frac{\text{mCherry RNA}_i}{\text{BFP RNA}_i} \times \frac{\text{BFP RNA}_{\text{sTRSVctl}}}{\text{mCherry RNA}_{\text{sTRSVctl}}} \times 100 \quad (6)$$

**Flow cytometry characterization of ribozyme switches**. HEK293T cells were seeded at 50,000 cells per well of a 24-well plate 24 h prior to transfection. Ligand was added 1 h before transfection. Transfection was performed using the Lipofectamine 2000 transfection reagent (Thermo Fisher Scientific) according to manufacturer's instructions, and 600 ng of plasmid were used in each transfection. Seventy-two hours later, cells were trypsinized and resuspended in DMEM (10% FBS, 1% penicillin/streptomycin) with no ligand. The final resuspended cells were assayed via flow cytometry on the MACSQuant flow cytometer, with forward scatter of gain 220 V, side scatter of gain 200 V, and using lasers with excitation wavelengths 561 nm, 405 nm, and 488 nm in channels Y2 (filter 615/20 nm; gain 313 V), V1 (filter 450/50 nm; gain 200 V), B1 (filter 525/50 nm; gain 200 V),

respectively. Analysis of the data was performed using a custom MATLAB script, available at https://github.com/jsxiang/FlowAnalysis, where cells are gated for viable and singlets, and transfected cells (BFP > $10^{2.7}$ fluorescence units are used in downstream analysis), (Supplementary Fig. 20). For each sequence tested in the pCS4076 vector, fluorescence intensity of *i* cells in the mCherry channel was normalized by the fluorescence intensity in the BFP channel to give the mean fluorescence ratio,

$$F_{\text{sample}} = \text{mean}\left(\frac{\text{mCherry}_i}{\text{BFP}_i}\right) \quad (7)$$

To determine the relative fluorescence unit, the mean fluorescence of the sample was normalized to that of the non-cleaving sTRSV ribozyme mutant sTRSVctl, and scaled by a factor of 100 as follows,

$$\text{RFU} = \frac{F_{\text{sample}}}{F_{\text{sTRSVctl}}} \times 100 \quad (8)$$

This normalization was performed to remove non-specific fluorescence effects due to the addition of ligands, and to standardize measurements across different experiments which may be subject to fluctuations in instrument gain. For sequences tested in the *eGFP* expressing pCS408 vector, all plasmids are co-transfected with a *mCherry* expression plasmid pCS407 at a 10:1 ratio to control for variability in transfection efficiency. The mean relative fluorescence values are determined as:

$$F_{\text{sample}} = \text{mean}\left(\frac{\text{GFP}_i}{\text{mCherry}_i}\right) \quad (9)$$

$$\text{RFU} = \frac{F_{\text{sample}}}{F_{\text{sTRSVctl}}} \times 100 \quad (10)$$

Statistical testing for significance in switching made use of unpaired, one-tailed *t*-tests implemented in MATLAB, and *p*-values for each set of with/without ligand comparisons were adjusted for multiple comparisons using the method of Benjamini and Hochberg using the MATLAB command mafdr ([list of *p*-values], 'BHFDR', 'true').

**Folate transporter overexpression**. Transporter sequences were obtained from mature mRNA transcript isoforms from National Center for Biotechnology Information (NCBI). Sequences were synthesized as qBlocks from QuintaraBio. One yeast folate transporter was included (SceFLR1), and was codon optimized for *Homo Sapiens* using the GeneArt software provided by Thermo Fisher Scientific. qBlock sequences (Supplementary Table 9) were PCR amplified using the Q5® High-Fidelity DNA Polymerase (New England Biolabs) using primers 5′-AGAC CCAAGCTGGCTAGC-3′ and 5′-AGGCTGATCAGCGGGTTTAA-3′, cleaned up using the DNA Clean & Concentrator kit (Zymo Research), and cloned into the BamHI XhoI sites of plasmid vector pcDNA/FRT via Gibson assembly. Gibson assembled products were transformed into One Shot™ TOP10 Chemically Competent *E. coli* (Thermo Fisher Scientific), plated on LB agar plates with carbenicillin (100 μg/ml), and incubated at 37 °C for 12–16 h. Individual colonies were sequence verified and inoculated in liquid LB media with carbenicillin (100 μg/ml). Plasmids were purified from cultures grown for 12–16 h using the Wizard® Plus SV Minipreps DNA Purification System kit (Promega) according to manufacturer's instructions.

Transporters were first screened for activity by co-transfecting the constructed plasmids at a 1:1 mass ratio with a FolUGAAG switch plasmid (pCS4123). The FolUGAAG switch was first identified as a potential switch candidate by having one of the lowest basal expression levels in RNA-Seq. Cells were assayed for fluorescent activity via the MACSQuant flow cytometer after 2 days of transfection and induction with 6 mM (6 R,S)-folinic acid. Transporter sequences that resulted in greater activation ratios of the FolUGAAG switch were identified as active.

Active transporters, namely *SLC19A1* and *SLC46A1*, were cloned into a lentiviral vector, pCDH-EF1α-MCS-(PGK-GFP) from System Biosciences (Palo Alto, CA, USA) via restriction sites XbaI/BamHI. The transporters were PCR amplified using the Q5® High-Fidelity DNA Polymerase (New England Biolabs) with primers 5′-TTCTTCCATTTCAGGTGTCGTGATCTAGAGCCACCATGGT GCCCTCCAGCCCA-3′ and 5′-TTGATTGTCGACGCGGCCGCGGATCCTCAC TGGTTCACATTCTGAACACC-3′ for SLC19A1, and 5′-TTCTTCCATTTCAG GTGTCGTGATCTAGAGCCACCATGGAGGGGAGCGCGAG-3′ and 5′-TTG ATTGTCGACGCGGCCGCGGATCCTCAGGGGCTCTGGGGAAA-3′ for *SLC46A1*, for Gibson assembly. Gibson assembly products were transformed into One Shot Stbl3™ Chemically Competent *E. coli* (Thermo Fisher Scientific) and plasmids were prepared using the Plasmid Plus Midi Kit (Qiagen) using the low-copy plasmid protocol to generate the lentiviral destination vectors pCS4134 and pCS4135. The destination vector was co-transfected with the packaging (pMO86) and envelope (pMO87) plasmids using a calcium phosphate transfection protocol[60]. Viruses were harvested after 2 days of transfection, and 1 ml of unconcentrated virus-containing media supplemented with 8 μg/ml of polybrene was used to transduce HEK293T cells, seeded at 200,000 cells in a 6-well plate the day before. The lentiviral construct contains a copGFP transduction marker, and cells were sorted for the top 5% brightest using FACS (~250,000 cells were

collected) on the FACSAria II cell sorter (BD Biosciences), using a 488 nm laser and 530/30 nm BP filter. The *SLC46A1*-transduced cell line resulted in greater switch induction and was thus used for screening and validating folinic acid switches expressed from pCS4076.

To enable switch characterization using a green fluorescence reporter, the copGFP marker for lentivirial transduction of HEK293T cells with the *SLC46A1* transporter was knocked out using CRISPR/Cas9. Three guide RNA sequences (TACGAGGCCGGCCGCGTGAT, CGCTGGGGTAGGTGCCGAAG, GTTCTC GTAGCCGCTGGGGT) targeting copGFP were designed using the CRISPR Design tool (crispr.mit.edu) and each cloned into a pX330-U6-Chimeric_BB-CBh-hSpCas9 vector (a gift from Feng Zhang, Addgene plasmid # 42230[61]). *SLC46A1* cells were transfected with the plasmids at 1:1:1 mass ratio and after 48 h were sorted for cells that expressed no copGFP on the FACSAria II cell sorter (BD Biosciences), using a 488 nm laser and 530/30 nm BP filter.

**Secondary structure folding prediction**. Folding predictions for folinic acid switch transcripts were carried out using the RNAstructure stochastic algorithm[50]. *mCherry*- and *eGFP*-linked folinic acid transcripts were both tested by using the transcript sequence, starting with the last 500 nucleotides of the fluorescent reporter and including the ribozyme switch and poly-A tail. One-thousand secondary structures were sampled for each design. The secondary structures were analyzed for correct folding of stem III of the hammerhead ribozyme. A score $\rho$, was assigned such that $\rho = \frac{\text{correctly folded structures}}{\text{all sampled structures}}$.

**Sequence and structural motif analysis**. To determine sequence motifs for each library with N4, N5, or N6 sequences, full-length sequences with >100 DNA read count coverage in each ligand and replicate condition were first aligned using the local alignment algorithm l-ins-i in the MAFFT alignment software package, version 7.309. The sequences were then prepared in MATLAB by binning into 14 activity bins of $\log_{10}(\text{RNA/DNA})$ from RNA-Seq results, where the first bin corresponds to sequences in the lowest 5th percentile of normalized RNA levels (two standard deviations below the mean), the last bin corresponds to sequences greater than the 68th percentile (one standard deviation above the mean), and the 12 intermediate bins are evenly divided. Sequence logo visualization of loop II sequences in each bin was performed using the ggseqlogo package in R/Rstudio.

Mutual information analysis provides structural information about two interacting positions (either directly or indirectly via interacting with a third base), where information (i.e., which base A, U, C, G) about one position $i$ reduces the uncertainty about another position $j$. Custom MATLAB script was written to perform mutual information analysis for loop II RNA sequences in each of the activity bins. From information theory, mutual information is given by,

$$\text{MI}(x, y) = \Sigma_{i,j} P_{i,j}(X, Y) \frac{P_{i,j}(X, Y)}{P_i(X) P_j(Y)} \quad (11)$$

where $X$ and $Y$ are base identities at positions $i$ and $j$, respectively, and $P_{i,j}(X, Y)$ is the observed weighted frequency of a pair of nucleotides, and $P_i(X)$ and $P_j(Y)$ are the independent weighted frequencies of individual nucleotides at their respective positions.

Pairwise contribution of base identities at each pair of two positions, $i$ and $j$, was determined using a custom script in MATLAB. Simply, the 5th or 10th percentile of $\log_{10}(\text{RNA/DNA})$ was computed for all sequences in each library harboring nucleotide $X$ at position $i$ and nucleotide $Y$ at position $j$, for all four possible bases, i.e., A, C, U, G.

To prepare the data for any downstream regression modeling, $\log_{10}(\text{RNA/DNA})$ values were standardized, i.e., sequences were translated to have mean = 0 and scaled to have standard deviation = 1. Sequence identities were also transformed into one-hot vectors. For a $m \times n$ vector of $m$ unique loop II sequences with $n$ bases (e.g., N5–6 library would have $n = 6$ after alignment), each element of sequence vector at position $i$ for $1 \leq i \leq m$ and $j$ for $1 \leq j \leq n$ is given by $X_{i \times j} = \{A, C, U, G\}$. The corresponding one-hot vector would be a $m \times 4n$ vector. Each element in the one-hot vector at position $i$ for $1 \leq i \leq m$ and $j$ for $1 \leq j \leq n$ is given by $X_{i \times (4(j-1)+k)} = \{0,1 \text{ if } X_k = N_k\}$, where $N_1 = A$, $N_2 = C$, $N_3 = U$, $N_4 = G$.

A boosting tree regression of one-hot encoded partial stem loop I and loop II sequence features to basal standardized $\log_{10}(\text{RNA/DNA})$ was performed using XGBoost[62] in Python 2.7 on theophylline, xanthine, folinic acid, cyclic di-GMP-II, and cyclic di-GMP I N5 libraries. The parameters used were 0.1 learning rate, max tree depth of 25, alpha of 10, using root mean squared estimator as the accuracy metric and 15-fold cross validation with a test:train split of 1:2 and 5 rounds of no improvement to early stopping. SHAP values are computed and visualized with the SHAP Python package[51].

The custom scripts used in this section are available at https://github.com/jsxiang/Ribozyme-SeqFun.

**Data denoising using AutoML**. Sequences with >100 read count were transformed to one-hot encoded 1's and 0's as described in the sequence and structural motifs identification section. Activity measures, namely RNA/DNA ratios, FACS-Seq relative fluorescence values, and activation ratios of both were standardized as described in the sequence and structural motifs identification section as separate, independent projects and different replicates. The models were trained to predict the different activity measures based on the input sequence information only, using the automated regression algorithm h2o.automl in the user-friendly java application H2O Flow and the R implementation of H2O version 3.20.0.6, which can be downloaded from https://www.h2o.ai/. Models were trained for 360 s on the one-hot encoded sequences and one activity measure each time, using the program's default parameters, including fivefold cross validation at 4:1 training:validation ratio to prevent over-fitting to learn general underlying sequence-activity relationships. The model was also set by default to stop after three iterations with no improvement in the default loss function. To generate the denoised data, converged models were then used to predict the RNA/DNA ratios and relative fluorescence values on the same one-hot encoded library sequences used as input.

The custom scripts are available at https://github.com/jsxiang/.

**Reporting summary**. Further information on research design is available in the Nature Research Reporting Summary linked to this article.

## Code availability

The authors declare that custom code used in the paper are available on Github, where the URL to specific repositories are as indicated at the relevant subsections in the Methods section.

## Data availability

The authors declare that the RNA-Seq, FACS-Seq and flow cytometry data supporting the findings of this study, i.e., the source data underlying Figs. 2–4 and Supplementary Figs. 9d, 12a, b, are available in the following Source Data files: Sequencing data are found on the NCBI Sequence Read Archive with accession numbers "SAMN12605057" for Theophylline Switch RNAseq, "SAMN12605058" for Folinic Acid and Xanthine RNAseq, "SAMN12605059" for Cyclic di-GMP RNAseq, "SAMN12605060" for Theophylline and cyclic di-GMP FACSseq, "SAMN12605061" for Xanthine FACSseq and "SAMN12605062" for Folinic Acid FACSseq. Data is also available under BioProject ID PRJNA560978. The following plasmids are deposited on Addgene: pCS4077 ("131738"), pCS4083 ("131739"), pCS4084 ("131740"), pCS4092 ("131741"), pCS4102 ("131742"), pCS4110 ("131743"), pCS4120 ("131744"), pCS4121 ("131745"), pCS4127 ("131746"), pCS4134 ("131747"). All other data are available from the authors upon request.

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

## Acknowledgements

We thank M.W. Covert, A. Cravens, E.J. Hayden, D. Kong, S. Li, S. Rudina, C. Schmidt, and F.E. Tan for valuable comments on the paper. Cell sorting for this project was performed on instruments in the Stanford Shared FACS Facility, with significant technical assistance from M. Weglarz and M. Bigos. Illumina MiSeq sequencing is performed at the Stanford Protein and Nucleic Acid Facility, with significant technical assistance from E. Zuo. We thank M. Mathur for assistance with lentivirus handling procedures, and C. Kim for assistance with FACS experiments. This work was supported by the National Institutes of Health (grant to C.D.S.), A*STAR (graduate fellowship to J.S.X.), NSERC (postdoctoral fellowship to M.M.), National Science Foundation (graduate fellowship to M.K. and P.D.), Howard Hughes Medical Institute (Gilliam graduate fellowship to M.K.), and Stanford REU and Stanford Undergraduate STEM Fellowship (M.H.).

## Author contributions

J.S.X. and C.D.S. conceived of the project and designed the experiments. J.S.X., M.K., and C.D.S. wrote the paper. J.S.X., M.K., P.D., and M.H. performed the experiments and analyzed the results. M.M. analyzed the results and edited the paper.

## Competing interests

C.D.S. is a co-founder and CEO of Antheia, Inc. and a co-founder of Chimera Bioengineering. The other authors declare no competing interests.
