## [Peer Review File · Nature Communications]

Reviewers' comments:

Reviewer #1 (Remarks to the Author):

This manuscript by Xiang et al. and Smolke describes the use of an in vivo selection process to identify RNA sequences that function as gene control devices in human cells. The authors highlight their use of FACS and RNA-Seq methods to screen through small populations of RNA constructs that join aptamer and self-cleaving ribozyme structures. Ultimately, they identify many RNA construct variants that exhibit gene expression increases of 2 to 9 fold when high concentrations of the matched ligand are added to the cell culture.

In general, the development of useful gene control devices that work in human cells is a very worthy pursuit, and the authors make this point apparent in various sections of their text. However, I strongly believe that the current work does not make major contributions to this area. Furthermore, the text is written in a manner that obscures the long history of RNA device engineering, and thereby inappropriately casts the current work as more pioneering than it is in reality. Finally, the text is overly wordy and filled with details that should be relegated to figures, tables, and supplementary materials. Therefore, in my comments below, I primarily note my most substantive concerns, rather than check the accuracy of all numbers, claimed facts, and conclusions.

Major concerns

1. On page 4, beginning with line 72, the authors acknowledge that there are “numerous proof-of-concepts studies” on synthetic ribozyme switches (originally called ‘allosteric ribozymes’, and also ‘aptazymes’). In the same sentence, the authors note the problems with the results of these past studies, including the fact that the switches “suffer from small dynamic ranges” and that a few “recurring set of aptamers” are used (which are points that have been made by others long ago). These are well-known problems of engineered allosteric ribozymes, and so there is no need to create more such RNAs unless these problems are to be overcome. Unfortunately, the authors add to the literature another large collection of exceedingly mediocre theophylline-dependent self-cleaving ribozymes based again on the hammerhead class. Although other aptamers are used to create similar allosteric ribozymes in this study, they have equivalently poor activities. By evaluating the outcome of the current study, readers will conclude that there has been no advance in the performances of engineered allosteric ribozymes. For example, no one will use an allosteric ribozyme in a human that exhibits a 9-fold increase in gene expression (at best, as most are 2 fold) when 1 mM theophylline (a toxic level to humans) is required as the inducer. So, the value of the current manuscript must be derived from other features of the work.

2. In parentheses in Point 1 above, I hint at another great weakness of the current manuscript. It is written in a way that completely ignores the past work on allosteric ribozymes, and even renames the device and its components. This has an effect of enhancing the appearance of the current work at the expense of others who have contributed to the field. Again on page 4, line 72, the authors rename allosteric ribozymes or aptazymes as “ribozyme switches”. On page 6, beginning on line 128, the authors use the terms “sensor domain” and “actuator domain” to replace the previously used aptamer and catalytic platform terms. This problem extends into the intellectual content regarding the challenges in this field. The authors incompletely describe some of the problems faced by RNA engineers, but ignore papers more clearly describing these problems that were published a decade or more ago. The current manuscript is already long and wordy, and could be easily shortened by citing this earlier work. Unfortunately, there is no novelty to be gained by the author’s discussion of the problems of the field, or by properly citing this past work.

3. Although the hammerhead ribozyme and theophylline, xanthine, c-di-GMP-I and c-di-GMP-II aptamers are not new to the allosteric ribozyme field, one could imagine joining them in unique ways. However, the construct designs in the current manuscript are quite similar to those used previously by others, although no mention of these past designs is made. Furthermore, despite the words used by the authors (e.g. massively-parallel, large datasets), the pool sizes are remarkably tiny compared to

most past studies employing selection or directed evolution methods. Even the Abstract provides a number (22,700) which appears to be the sum of the sequences tested for all constructs used in the study. By contrast, some test-tube evolution studies on allosteric ribozymes have used pool sizes that included about a trillion different molecules. The authors tout the benefits of doing screening in cells, but pool size is not one of them! Therefore, I do not think that novelty can be derived from the author's construct designs. Likewise, FACS and RNA-seq methods have also been used previously for identifying RNA switches that function in cells, and so this part lacks technical novelty as well.

4. The authors do seem to apply many statistical analyses with their datasets, and describe all these numbers in far too great detail in the text. However, this writing style only contributes to obscuring the fact that the switches are remarkably poor, and that the study lacks impact and novelty. Unfortunately, simplifying the text by shifting this unnecessary detail to figures, tables, or supplementary data will not help the author's case for publication. Simply put, the authors have used relatively routine methods to create additional theophylline-dependent RNA switches that no one will use in cells.

Reviewer #2 (Remarks to the Author):

In this study, the authors develop an RNA-seq-based assay for screening the activity of small molecule-responsive RNAs in mammalian cells, and validate this against other measurements. A key advantage of this approach is the use of quantitative sequencing as a basis for normalizing libraries introduced by transient transfection, greatly improving the speed and (potentially) reducing the noise associated with such a screen. This assay was used to identify new ribozyme switches for detecting small molecule ligands of interest. A few points to be addressed are noted below, but overall this is a data-rich study with careful and interesting analysis (especially in Figure 5), and it would be a valuable contribution to the literature on the topics of ribozyme engineering and high throughput generation of new "parts" for synthetic biology.

Major comments

- There are several items relating to statistical tests and correlations that should be addressed, as delineated in comments below on figures and captions. In general, more precise explanations for the statistical tests should be provided in figure captions, false discovery rate correction should be addressed by implementing the appropriate tests, and the asterisks denoting statistical significance between conditions and any description of these results in the text should be updated as needed based on this correction.
- Page 4: could the authors provide more explanation (either in the introduction or elsewhere in the main text) as to how the activity of ribozyme switches and their regulation of gene expression are expected to differ between cell-free, non-mammalian, and mammalian systems? This is mentioned briefly but could be expanded. Why is a specific RNA-ligand pair an insufficient determinant for whether a high-performing switch in one context will exhibit activity in another context? Does self-cleavage not occur at the same rate or to the same extent? Or, are differences in outcomes attributed more to events that occur downstream of the self-cleavage event?

Other comments

- Page 3, line 50: the broad claim conveying RNAs necessarily have greater versatility is confusing, and it can be rephrased or removed. It is unclear how this type of claim could be demonstrated, and

such statements are not needed to motivate the study, which is already compelling. (RNAs are better suited than other biomolecules for some purposes, and other biomolecules are better suited than RNAs for other purposes.)

- page 3 line 53/54: therapeutics isare, due to itstheir
- page 3 line 60: control gene  control of gene
- page 4 line 68: advantages over what?
- page 4 line 76: are designed can be designed
- Page 7, lines 145–148: would additional sources of variation, besides the DNA synthesis step, include cell-to-cell variation in plasmid uptake and/or plasmid-to-plasmid variation in uptake during transfection? If multiple variants in the library can be delivered to the same cell, is there any way this potentially affects the assay, and if so, can this aspect be explained in the main text? It is clear that the normalization employed (normalization to total DNA reads) is conservative, but these factors warrant discussion.
- Page 9 line 189: can you provide rationale as to why the DNA abundance was quantified post-transfection/recovery rather than by direct sequencing of the plasmid DNA library prior to transfection? What additional sources of bias are corrected for using this strategy?
- Page 11, lines 234–235: it would be appropriate to clarify that the Poisson-like process of lentiviral integration does not “ensure” single-copy integration, but rather that choosing a low MOI causes a large percentage of cells to undergo at most one integration.
- Page 12, line 259: “The data suggest”.
- Page 12, line 264: the term “significantly” should be reserved for if there is an accompanying statistical test.
- Page 13, line 290: check grammar.
- Page 13, line 291–292: the statement beginning with “thereby” seems like a conclusion that can be made, but only at the end of the section corresponding to Figure 4.
- At some point, even if only parenthetically, it could be helpful to provide more context for the observed ligand-induced fold differences in signal from flow cytometry assays for validating RNAs, so these values can be better interpreted in light of an expected or typical range for these systems.
- Page 16, line 366: please be careful with wording; the dynamic ranges measured by flow cytometry are not “over 2 fold”.
- Page 17, line 380–381: is there more of a basis for this speculation? This seems to be mentioned but only in the discussion.
- Page 17, lines 385, 387: the assay does not seem to directly measure which variants are the “fastest cleaving”, per se. Please rephrase.
- Page 19, line 432, and other instances: the 5th percentile, or top 5th percentile? Please clarify.

Figures and captions

- Fig. 1a: to some readers, the Pac-men may convey a mechanism involving enzyme recruitment. Consider adjusting the cartoon to show only self-cleavage, or indicate in the caption what the Pac-men represent.
- Fig. 1c: the workflow diagram seems to convey that the plasmid DNA and the total RNA are extracted from the same set of transfected cells, but page 9 lines 189–190 contains a statement to the contrary. Please adjust the figure as needed.
- In the main figures and supplementary figures, whenever an R^2 value is shown for a plot with log-scaled axes, it is unclear if R^2 was calculated based on log-transformed or non-transformed values. Is it consistently one or the other, or does it vary? Please clarify in the figure captions.
- Fig. 2d: the statistical tests need clarification. Which conditions are compared, is it 0 vs. 5 mM? Are these one-tailed tests? Was a correction for false discovery rate, such as the Benjamini-Hochberg procedure, applied for the set of tests in this panel?

- Fig. 2e–f: is the lowest y-axis tick value 3 or 3.2?
- Fig. 4j–l: which conditions are compared, is it 0 vs. 12.5 mM? Was a correction for false discovery rate applied?
- Fig. 4 and the main text: needs consistency in use of eGFP, not GFP.

Reviewer #3 (Remarks to the Author):

The authors present a well-written manuscript describing a substantial body of work to develop a method for indirectly assessing the activity of engineered riboswitch designs expressed in mammalian cells that will certainly be of interest to the field. I am particularly impressed by their extensive validation using control ribozymes and libraries. In general, the authors claims are supported by the data presented and the methods are described in sufficient detail for others to reproduce their work. I would support publication in *Nature Communications* provided that the authors can address the comments below.

This work builds upon existing methods previously described by the Smolke lab and, as the authors acknowledge, the Yokobayashi lab (UC Davis). As in previous methods, RNA-seq is used to quantify levels of transcripts containing self-cleaving aptazyme designs, and the novelty here is the speed of the method and the use of DNA sequence information to normalize RNA data in order to reduce errors arising from inter alia differences in plasmid copy numbers and/or differences in amplification during the workup.

An overall comment is that whilst the method indeed appears to be sound for identifying improved designs present in the libraries, the dynamic range of the best variants is still relatively modest; no ribozyme described here has a dynamic range greater than one order of magnitude. Could the authors comment on whether they feel that this is limited by the designs investigated, or these particular aptamers / ribozyme, or other factors? The method is restricted to very small library sizes (compared with, for example, SELEX libraries) - given this I would suspect that the method may be limited to fine-tuning new designs rather than an extensive exploration of sequence space that may contain more sophisticated aptazymes.

Specific comments:

1. The authors state that a similar method described by the Yokobayashi lab (Nomura et al. 2017 Chem. Comm.) is limited to screening ribozyme designs of the same length (page 5, line 106). However, as I understand it, whilst the different aptazyme library designs differ in length, within each library the variants examined here are also of the same length. Have the authors examined whether their method indeed allows a library containing length variants (i.e. with indels) to be screened?
2. The authors state that DNA plasmids were extracted from separately transfected cells than those used for the RNA-Seq workup. As transfection efficiency can vary between batches of cells, does this not unnecessarily introduce a source of error? Would it not have been better to transfect twice as many cells then use half for DNA and half for RNA?
3. In figure 2d, the change in mCherry fluorescence is used to report activation of a range of theophylline-binding riboswitch variants identified using the RNA-Seq screen. The level of activation at 1mM and 5mM appears to be very similar, indicating that maximum activation is already achieved at

1mM. It would have been reassuring that the switches are working as intended to see a dose response with activation scaling with theophylline concentration. This would be particularly relevant if, for example, a riboswitch is being engineered to provide a quantitative response to a ligand of interest. Do the authors have data for lower concentrations of theophylline? If so do these indeed show a dose response?

4. Figure 3b is not referred to in the main text. Presumably a reference to this panel should appear on Page 11 line 253.

5. Page 15. The authors describe how the folinic acid riboswitch library was incompatible with the coding sequence of mCherry ostensibly due to complementarity, leading to misfolding and inactivation of the ribozyme in this library, and hence high basal expression of the reporter. However, in supplementary note 3 (and referring to data in supplementary figure 8) the authors note that bulk cleavage of this library was comparable to other libraries *in vitro*, which would not be the case if misfolding occurred in this assay. Presumably the mCherry RNA sequence alone is insufficient to cause misfolding (as the authors claim in the main text, page 15 lines 345-8), and other factors affect the folding inside cells – this should be noted in the main text.

6. I found the section dealing with conservation in sequence and structure of ribozymes quite difficult to follow. In particular, given that (as the authors acknowledge on page 19 lines 438-441) interactions between the loop residues varied in the libraries and the linked aptamers is likely to be specific for the aptamers (and the ribozyme) used in this study, I am unsure how the authors' findings "could be used to inform future rational design efforts" (Abstract lines 30-32) in any concrete way for other aptamers and ribozymes – could the authors be more specific in this claim?

7. As the authors acknowledge, the FACS-Seq data appears to be very noisy. I am unfamiliar with the method used to improve the signal to noise so cannot comment on whether this is appropriate.

8. In Supplementary figure 5, some of the plots appear to show two fit lines –are these a fit of each replicate data set, or of the raw vs denoised data for both replicates? Why do some panels only show one fit?

9. In Supplementary figure 8, the legend suggests that the pairs of reactions shown in the gel(s) are -/+ ligand – this should be indicated on the figure. The reaction conditions and timing should be indicated in the figure legend.

Response to Reviewer Comments

We would like to thank the reviewers for their constructive comments and helpful suggestions for improving our manuscript. The comments were very useful in helping us to strengthen the manuscript and clarify the presentation of our work. In the sections below, we address each comment made by the reviewers.

Reviewer #1 (Remarks to the Author):

This manuscript by Xiang et al. and Smolke describes the use of an in vivo selection process to identify RNA sequences that function as gene control devices in human cells. The authors highlight their use of FACS and RNA-Seq methods to screen through small populations of RNA constructs that join aptamer and self-cleaving ribozyme structures. Ultimately, they identify many RNA construct variants that exhibit gene expression increases of 2 to 9 fold when high concentrations of the matched ligand are added to the cell culture.

In general, the development of useful gene control devices that work in human cells is a very worthy pursuit, and the authors make this point apparent in various sections of their text. However, I strongly believe that the current work does not make major contributions to this area. Furthermore, the text is written in a manner that obscures the long history of RNA device engineering, and thereby inappropriately casts the current work as more pioneering than it is in reality. Finally, the text is overly wordy and filled with details that should be relegated to figures, tables, and supplementary materials. Therefore, in my comments below, I primarily note my most substantive concerns, rather than check the accuracy of all numbers, claimed facts, and conclusions.

We thank the reviewer for the comments and feedback on the manuscript, and will work with the editor to edit for length and conciseness.

Major concerns

1. On page 4, beginning with line 72, the authors acknowledge that there are “numerous proof-of-concepts studies” on synthetic ribozyme switches (originally called ‘allosteric ribozymes’, and also ‘aptazymes’). In the same sentence, the authors note the problems with the results of these past studies, including the fact that the switches “suffer from small dynamic ranges” and that a few “recurring set of aptamers” are used (which are points that have been made by others long ago). These are well-known problems of engineered allosteric ribozymes, and so there is no need to create more such RNAs unless these problems are to be overcome. Unfortunately, the authors add to the literature another large collection of exceedingly mediocre theophylline-dependent self-cleaving ribozymes based again on the hammerhead class. Although other aptamers are used to create similar allosteric ribozymes in this study, they have equivalently poor activities. By evaluating the outcome of the current study, readers will conclude that there has been no advance in the performances of engineered allosteric ribozymes. For example, no one will use an allosteric ribozyme in a human that exhibits a 9-fold increase in gene expression (at best, as most are 2 fold) when 1 mM theophylline (a toxic level to humans) is required as the inducer. So, the value of the current manuscript must be derived from other features of the work.

We appreciate the reviewer’s comments, but respectfully disagree about the significance of our work. The significance stems from developing a robust and high-throughput method to systematically and quantitatively assess *in vivo* characteristics of ribozyme switch performance in controlling gene expression in mammalian cells, while providing quantitative insight into the sequence function space of RNA-based switch performance in living cells. This method can be used more broadly by the research

community to design and develop cis-acting RNA gene control devices that modulate transcript levels more rapidly and successfully

Our work represents a significant advancement in the field for engineering ribozyme ‘ON’ switches with 200% to 300% improvement in dynamic range from currently reported constructs in the literature. The field has demonstrated ribozyme switches (or allosteric ribozymes or aptazymes) taking the form of ‘ON’ or ‘OFF’ switches. In the former, the presence of ligand inhibits self-cleavage, which results in induced gene expression; in the latter, the presence of ligand is needed to facilitate self-cleavage and result in reduced gene expression. While there have been reports of ‘OFF’ switches exhibiting dynamic ranges of greater than 10-fold in down-regulating luciferase reporter expression¹⁻³, the design principles for engineering these ‘OFF’ switches fail to translate to ‘ON’ switches with >10-fold activation ratio^{3,4}. The largest dynamic range reported for an ‘ON’ switch in mammalian cells is 3.5 fold using a tetracycline-responsive switch to regulate fluorescent reporter expression⁵, which the authors show corresponds to an 8.7-fold activation ratio using luciferase as a reporter. It should be noted that luciferase assays are enzymatic and represents a non-linear and amplified measure of protein expression levels that are not directly comparable to those measured through fluorescent reporter proteins. A recent study demonstrated a guanine-responsive ‘ON’ switch with an activation ratio of 4.0 fold in luciferase assays³, but no fluorescent reporter based activation ratio was reported. Theophylline switches have been reported to exhibit activation ratios of ~2-3 fold^{6,7} in fluorescent protein and luciferase assays in mammalian cells, and xanthine, folinic acid, and c-di-GMP switches have yet to be demonstrated in mammalian cells prior to our work. Based on the differences in mechanism and design principles required, comparing fold change from an ‘OFF’ switch to that from an ‘ON’ switch is not relevant. We agree there is much to learn from the engineering literature and history of ‘OFF’ switches and allosteric ribozymes, but it has not been sufficient to enable the research community to engineer ‘ON’ switches with similar dynamic ranges. Therefore, our method enabled us to identify ribozyme switches that exhibit two to three times improvement in the dynamic range of ‘ON’ switches reported and new ligand-responsive switches in mammalian cells, representing a significant advance for the field.

We report activation ratios at high ligand concentrations to demonstrate the full dynamic range of each switch, but the actual fold change appropriate for a particular system will depend on the requirements of each application. One advantage of ribozyme switches over bi-stable transcription factor inducible systems is associated with their dose response characteristic. We now include a dose response curve for three theophylline switches in Figure 2e to show different levels of activation depending on ligand concentration. While serum levels of 1 mM theophylline are toxic in humans, more typical serum levels of theophylline at around 20 µg/ml or 111 µM show a 2-fold activation using our theophylline switches (e.g., TheoCAUAA, TheoAAAAA) which is sufficient for certain gene regulation applications^{6,8} that often do not require or desire high expression of the target proteins. Depending on the application, the research community has shown that gene expression changes at one node can be amplified through downstream pathways to achieve desired outcome. Examples include ribozyme and miRNA switches exhibiting ~2-3 fold activation ratios achieving over ten fold amplification of cell count^{6,8}, and over 40-fold induction or 1000-fold reduction in viral titer⁷.

2. In parentheses in Point 1 above, I hint at another great weakness of the current manuscript. It is written in a way that completely ignores the past work on allosteric ribozymes, and even renames the device and its components. This has an effect of enhancing the appearance of the current work at the expense of others who have contributed to the field. Again on page 4, line 72, the authors rename allosteric ribozymes or aptazymes as “ribozyme switches”. On page 6, beginning on line 128, the authors use the terms “sensor domain” and “actuator domain” to replace the previously used aptamer and catalytic platform terms. This problem extends into the intellectual content regarding the challenges in this field. The authors incompletely describe some of the problems faced by RNA engineers, but ignore papers more clearly describing these problems that were published a decade or more ago. The current manuscript is already long and wordy, and could be easily shortened by citing

this earlier work. Unfortunately, there is no novelty to be gained by the author's discussion of the problems of the field, or by properly citing this past work.

We thank the reviewer for the comment. We have strived to include references and discussion of work in the field that appropriately put our work in context, particularly as it relates to research that has used ligand-responsive ribozymes as gene control elements in eukaryotic systems (in the main text, references 25 to 28, 31 to 37, 39, 40, 44, and 47). If we have missed particular work that is important for providing that context, we would be happy to include references to those papers. We acknowledge that we are building on a large and rich body of work, starting from early efforts on engineering *in vitro* aptazymes and catalytic ribozymes. There are unique challenges the community faces in applying ribozymes as gene control elements, and thus we framed our introduction to focus on more recent advances in this space.

We also appreciate the reviewer's comments on terminology. We revised the text to mention that ribozyme switches have also been referred to as allosteric ribozymes or aptazymes. However, our manuscript is written from the perspective of providing a synthetic biology framework to the characterization of genetic devices and parts. The synthetic biology community looks to provide abstraction frameworks, where systems and devices can be composed of parts that are defined by function. Thus, as a broad framing, gene control devices are generally composed of sensor and actuator parts or domains. RNA-based gene control devices (or switches) in this framing are composed of a sensor domain (aptamer) and actuator domain (ribozyme). Since our focus is on engineering ligand-responsive ribozymes as gene control switches within a cellular context, we believe this framing is more appropriate and reaches a broader community of researchers generally interested in the engineering of biological systems.

3. Although the hammerhead ribozyme and the theophylline, xanthine, c-di-GMP-I and c-di-GMP-II aptamers are not new to the allosteric ribozyme field, one could imagine joining them in unique ways. However, the construct designs in the current manuscript are quite similar to those used previously by others, although no mention of these past designs is made.

We thank the reviewer for the comment. We would like to clarify that for the purpose of method development, we chose to base our switch library designs on the well studied hammerhead ribozyme class of RNA switches. We respectfully acknowledge that our designs are inspired by similar previous work that have shown the most promise for engineering optimal 'ON' switches, but which also critically needed a thorough and systematic exploration of sequence space for identifying the rare sequences in a library pool with the largest dynamic range for mammalian cell systems. We revised the text to more clearly acknowledge that ribozyme switches (or allosteric ribozymes) for c-di-GMP, xanthine, and folinic acid have been engineered in cell-free systems⁹ and microbial systems^{10,11}, but they have yet to be shown to demonstrate switching activity in mammalian cells. Theophylline 'ON' switches have also only been reported to show around 2-3 fold activation ratio in mammalian cells^{6,7}. Nevertheless, we would gladly cite any other relevant work that we have inadvertently missed.

Furthermore, despite the words used by the authors (e.g. massively-parallel, large datasets), the pool sizes are remarkably tiny compared to most past studies employing selection or directed evolution methods. Even the Abstract provides a number (22,700) which appears to be the sum of the sequences tested for all constructs used in the study. By contrast, some test-tube evolution studies on allosteric ribozymes have used pool sizes that included about a trillion different molecules. The authors tout the benefits of doing screening in cells, but pool size is not one of them! Therefore, I do not think that novelty can be derived from the author's construct designs. Likewise, FACS and RNA-seq methods have also been used previously for identifying RNA switches that function in cells, and so this part lacks technical novelty as well.

We thank the reviewer for the comment. Test tube evolution has been shown to access large sequence spaces of up to 10^{15} which the field deems necessary for aptamer discovery in SELEX experiments, and evolving ribozyme catalysis *de novo*, but are limited to cell-free systems which may not capture differences between *in vitro* and cellular systems that impact ribozyme switch function. Examples of such factors include differences in transcription rate, self-cleavage rates, cell-dependent translation and RNA degradation rates. These SELEX experiments are also evolution selection studies which aimed to enrich the library down to individual sequence clones for a handful of characterizations or up to a million for deep sequencing enrichment analysis¹², and are not capable of simultaneously measuring the activity of trillions of molecules. Currently, our method is based on simultaneously screening a N5-6 library with 5,120 member sequence diversity in each well of 6-well plates, and sequencing on a MiSeq format Illumina sequencer. To the best of our knowledge, mammalian cell-based RNA switch optimization has only been performed manually or screened in microbial systems, which also bear differences from mammalian cell systems. On the other hand, our methods are readily scalable to an order of magnitude greater than the numbers that we reported. Transfection experiments can be scaled up to 15-cm plates for at least $145 \text{ cm}^2 / 9.6 \text{ cm}^2 \approx 15$ times greater sequence space or 77,000 sequence variants using conservative estimates. Presently, Illumina sequencer throughput also reaches over a billion reads for only roughly double the cost compared to ~ 20 million reads from MiSeq, which again makes scaling up feasible. We did not choose to screen larger libraries because our previous work on this hammerhead class of ribozyme switches enabled us to narrow down the sequence search space to N5-6 to identify optimal candidates. For future cell-based screening efforts exploring newer ribozyme classes and requiring larger sequence space, our current methodology will be highly applicable.

Regarding the novelty of our reported methods, RNA-Seq has widespread use in a large variety of applications to measure differential gene expression, but it has not been developed to measure mRNA expression levels with quantitative accuracy as regulated by ribozyme switches. Our insight of DNA library abundance accounting for the majority of variation in a sequencing read count based assay of library pools also enabled us to achieve greater accuracy than using RNA reads alone (Figure 3f and 3g; Supplementary Figure 1). In addition, FACS-Seq in mammalian cells have yet to be developed for screening synthetic RNA devices, and presents numerous new challenges compared to the FACS-Seq in microbial systems referred to by the reviewer. These challenges are mainly resulting from the need to use lentiviral transductions, which we elaborate on in our supplementary note regarding noise. We developed both methods and further made systematic side-by-side comparison to conclude that RNA-Seq with noise reduction by normalizing DNA pool abundance variance is a simple, accurate, and robust method to facilitate future engineering efforts.

4. The authors do seem to apply many statistical analyses with their datasets, and describe all these numbers in far too great detail in the text. However, this writing style only contributes to obscuring the fact that the switches are remarkably poor, and that the study lacks impact and novelty. Unfortunately, simplifying the text by shifting this unnecessary detail to figures, tables, or supplementary data will not help the author's case for publication. Simply put, the authors have used relatively routine methods to create additional theophylline-dependent RNA switches that no one will use in cells.

We thank the reviewer for acknowledging the highly detailed nature of our statistical analyses, which is particularly important with these quantitative assays based on NGS. We value the presentation of transparent and clear evidence to substantiate the systematic and quantitative claims about the methods to encourage reuse and improve reproducibility across the community. We will work with the editor to edit for length and conciseness as appropriate for journal requirements.

We developed RNA-Seq for rapidly assaying ribozyme switches in a quantitative and high-throughput fashion, which represents a substantial advance for accelerating the process of screening RNA switches directly in mammalian cells. Using these newly developed methods, we engineered RNA switches to four ligands, which include an over 300% improvement in activation ratio of the best in class

theophylline switch over those previously reported and significantly expanded existing switches available to new ligands xanthine, folinic acid, and c-di-GMP in mammalian cell systems.

Reviewer #2 (Remarks to the Author):

In this study, the authors develop an RNA-seq-based assay for screening the activity of small molecule-responsive RNAs in mammalian cells, and validate this against other measurements. A key advantage of this approach is the use of quantitative sequencing as a basis for normalizing libraries introduced by transient transfection, greatly improving the speed and (potentially) reducing the noise associated with such a screen. This assay was used to identify new ribozyme switches for detecting small molecule ligands of interest. A few points to be addressed are noted below, but overall this is a data-rich study with careful and interesting analysis (especially in Figure 5), and it would be a valuable contribution to the literature on the topics of ribozyme engineering and high throughput generation of new “parts” for synthetic biology.

We thank the reviewer for their supportive remarks and suggestions.

Major comments

- *There are several items relating to statistical tests and correlations that should be addressed, as delineated in comments below on figures and captions. In general, more precise explanations for the statistical tests should be provided in figure captions, false discovery rate correction should be addressed by implementing the appropriate tests, and the asterisks denoting statistical significance between conditions and any description of these results in the text should be updated as needed based on this correction.*

We thank the reviewer for these suggestions and have revised the manuscript to provide more clear explanations for statistical tests in the text and figure captions. In addition, the Benjamini-Hochberg false discovery rate method has now been applied to all RNA-Seq, FACS-Seq and flow cytometry assay tests for significance in switching and the manuscript text is updated accordingly based on this analysis. Specifically, the Benjamini-Hochberg adjusted p-values are now indicated as asterisks, in replacement of p-values, in the flow cytometry result figures. We edited the text to mention that the previously used p-value < 1e-125 criteria for identifying sequences with significant switching in FACS-Seq corresponds to Benjamini-Hochberg adjusted p < 1.5e-125. Accordingly, all source data tables for RNA-Seq, FACS-Seq, and validation are also updated with the Benjamini-Hochberg adjusted p-values.

- *Page 4: could the authors provide more explanation (either in the introduction or elsewhere in the main text) as to how the activity of ribozyme switches and their regulation of gene expression are expected to differ between cell-free, non-mammalian, and mammalian systems? This is mentioned briefly but could be expanded. Why is a specific RNA-ligand pair an insufficient determinant for whether a high-performing switch in one context will exhibit activity in another context? Does self-cleavage not occur at the same rate or to the same extent? Or, are differences in outcomes attributed more to events that occur downstream of the self-cleavage event?*

We thank the reviewer for the comment. There are a number of differences between cell-free, non-mammalian, and mammalian systems that can impact the activity of ribozyme switches as measured by the ribozyme self-cleavage rate, mRNA levels, or protein levels. Some of these differences can impact the self-cleavage rate, including: i. differences in transcription rates between all three systems which may result in differences in kinetic folding¹³ and ii. differences in intracellular Mg²⁺ concentration, which affects hammerhead ribozyme cleavage rates. Other differences impact events that occur downstream of the cleavage event, including, i. differences in whether translation occurs co-transcriptionally or only after the transcribed RNA is exported from the nucleus, ii. differences in endogenous RNA binding proteins

that regulate mRNA degradation rates downstream of the self-cleavage event, which would impact mRNA half life, iii. differences in ligand permeability which would impact intracellular ligand concentration. These factors are extrinsic to RNA-ligand pair properties and can result in a high-performing switch in one context not translating to another context. We have modified the manuscript text to include more explanation of differences between cell-free, non-mammalian, and mammalian systems that may impact the measured activity of ribozyme switches.

Other comments

• *Page 3, line 50: the broad claim conveying RNAs necessarily have greater versatility is confusing, and it can be rephrased or removed. It is unclear how this type of claim could be demonstrated, and such statements are not needed to motivate the study, which is already compelling. (RNAs are better suited than other biomolecules for some purposes, and other biomolecules are better suited than RNAs for other purposes.)*

We thank the reviewer for this suggestion and have deleted the versatility claim in the revised manuscript text.

• *page 3 line 53/54: therapeutics isare, due to itstheir*

We have revised the manuscript text as suggested by the reviewer.

• *page 3 line 60: control gene  control of gene*

We have revised the manuscript text as suggested by the reviewer.

• *page 4 line 68: advantages over what?*

We thank the reviewer for this suggestion and have revised the text to clarify that the advantages cited in this sentence are compared with inducible expression systems requiring heterologous regulatory proteins such as transcription factor-based systems.

• *page 4 line 76: are designed can be designed*

We have revised the manuscript text as suggested by the reviewer.

• *Page 7, lines 145–148: would additional sources of variation, besides the DNA synthesis step, include cell-to-cell variation in plasmid uptake and/or plasmid-to-plasmid variation in uptake during transfection? If multiple variants in the library can be delivered to the same cell, is there any way this potentially affects the assay, and if so, can this aspect be explained in the main text? It is clear that the normalization employed (normalization to total DNA reads) is conservative, but these factors warrant discussion.*

We thank the reviewer for the comment. Additional sources of variation can arise during any experimental stage of the workflow, if coverage at any of these steps falls below ~100-fold for each molecule in the library, from library construction to transfection efficiency, and finally sequencing read coverage. We agree that if transfection efficiency is low, there may be cell-to-cell variation that could affect the results, such as sequences in the library not being represented, or the relative abundance of each library member not being preserved due to sampling noise. Therefore, we routinely check for transfection efficiency by fluorescence microscopy and include extra transfection wells to quantify transfection efficiencies by flow cytometry (typically observed ~20-40%). If our transfection efficiency is low (below ~5%) we do not go forward with analysis. By qPCR quantification of the reverse transcribed cDNA from

the total isolated RNA, we typically observe 10^4 to 10^6 fold representation of each library member from the recovered RNA, i.e., for a N5-6 library with 5,120 diversity, at least 5120×10^4 RNA molecules containing the switches were recovered. This level of sequence coverage allows for robust statistical analysis of these sequences based on our experimental results. We have not observed that multiple variants being delivered to the same cell would affect the assay, as there is a strong correlation between the RNA-Seq derived normalized RNA levels, where each library member is transfected alongside the rest of the library pool, versus the qPCR validation results as shown in Supplementary Figure 2, or the flow cytometry validation results, where only one ribozyme sequence was transfected in each sample and independently assayed.

We revised the text to mention that additional sources of bias include sequencing depth, *E. coli* plasmid transformation rate, or insufficient transfection efficiency, that this type of bias can be corrected by ensuring approximately at least a 100-1,000 fold library coverage at every experimental stage.

- *Page 9 line 189: can you provide rationale as to why the DNA abundance was quantified post-transfection/recovery rather than by direct sequencing of the plasmid DNA library prior to transfection? What additional sources of bias are corrected for using this strategy?*

We thank the reviewer for the comment. We chose to analyze the DNA abundance post-transfection in order to correct for any bias possibly introduced in the transfection process, e.g., cell-to-cell variation in plasmid uptake. This is to ensure that we only counted sequences that were successfully transfected.

- *Page 11, lines 234–235: it would be appropriate to clarify that the Poisson-like process of lentiviral integration does not “ensure” single-copy integration, but rather that choosing a low MOI causes a large percentage of cells to undergo at most one integration.*

We have modified the manuscript text to clarify that using a low MOI in the lentiviral transduction results in a large percentage of cells undergoing at most one integration.

- *Page 12, line 259: “The data suggest”.*

We have revised the manuscript text as suggested by the reviewer.

- *Page 12, line 264: the term “significantly” should be reserved for if there is an accompanying statistical test.*

We have removed “significantly” from this sentence as suggested by the reviewer.

- *Page 13, line 290: check grammar.*
- *Page 13, line 291–292: the statement beginning with “thereby” seems like a conclusion that can be made, but only at the end of the section corresponding to Figure 4.*

We thank the reviewer for the suggestion and revised the manuscript text accordingly to remove the use of “thereby”.

- *At some point, even if only parenthetically, it could be helpful to provide more context for the observed ligand-induced fold differences in signal from flow cytometry assays for validating RNAs, so these values can be better interpreted in light of an expected or typical range for these systems.*

We thank the reviewer for this suggestion and now include a mention in the introduction that the current highest activation ratio of an ON switch is 3.5 fold in fluorescent reporter assays in mammalian cells.

• Page 16, line 366: please be careful with wording; the dynamic ranges measured by flow cytometry are not “over 2 fold”.

We thank the reviewer for this comment and have modified the main text to say “up to 2-fold.”

• Page 17, line 380–381: is there more of a basis for this speculation? This seems to be mentioned but only in the discussion.

We thank the reviewer for this comment. To the best of our knowledge, there is no report of c-di-GMP penetration or concentration inside human cells after incubation. We base our speculation on similar observations in engineering folinic acid switches. Before overexpression of the SLC46A1 folate transporter, the folinic acid switches exhibited low relative GFP expression suggesting much room for induced GFP expression, but these switches were unable to reach full induction when the switch transfected cells were incubated with folinic acid ligand. We reasoned that overexpression of the SLC46A1 folate transporter was needed to increase the intracellular concentration of folinic acid to induce the switches. Like folinic acid switches, the c-di-GMP II switches had low basal expression, which saturated even around just 400 μ M of the c-di-GMP ligand. However, since c-di-GMP isn't a natural ligand of human cells unlike folates, transport issues may be hindering cell penetration of the ligand. We revised the text to mention that the basis of our speculation is based on comparison with folinic acid switches that did not exhibit closer to full induction until overexpression of the SLC46A1 transporter.

• Page 17, lines 385, 387: the assay does not seem to directly measure which variants are the “fastest cleaving”, per se. Please rephrase.

We thank the reviewer for this suggestion. We have revised the text to state that we are measuring sequences with low basal expression rather than fastest cleaving.

• Page 19, line 432, and other instances: the 5th percentile, or top 5th percentile? Please clarify.

We thank the reviewer for this suggestion and have revised the text to state lowest 5th percentile.

Figures and captions

• Fig. 1a: to some readers, the Pac-men may convey a mechanism involving enzyme recruitment. Consider adjusting the cartoon to show only self-cleavage, or indicate in the caption what the Pac-men represent.

We have revised Figure 1a and Supplementary Figure 4a to remove the Pac-men.

• Fig. 1c: the workflow diagram seems to convey that the plasmid DNA and the total RNA are extracted from the same set of transfected cells, but page 9 lines 189–190 contains a statement to the contrary. Please adjust the figure as needed.

We thank the reviewer for this suggestion. We have revised Figure 1 to reflect that plasmid DNA and total RNA are extracted from separate sets of transfected cells.

• In the main figures and supplementary figures, whenever an R^2 value is shown for a plot with log-scaled axes, it is unclear if R^2 was calculated based on log-transformed or non-transformed values. Is it consistently one or the other, or does it vary? Please clarify in the figure captions.

We thank the reviewer for this comment. We have revised all relevant figure captions to clarify that R^2 was calculated based on log-transformed values for comparing normalized RNA levels, qPCR values, and

FACS-seq and flow cytometry-derived relative fluorescence values. In all other cases, R^2 was calculated based on non-log-transformed values.

- *Fig. 2d: the statistical tests need clarification. Which conditions are compared, is it 0 vs. 5 mM? Are these one-tailed tests? Was a correction for false discovery rate, such as the Benjamini-Hochberg procedure, applied for the set of tests in this panel?*

We thank the reviewer for this comment. We have revised the figure caption to clarify the ligand conditions that are being compared in this figure and the statistical tests. In addition, the Benjamini-Hochberg adjusted p-values are now reported for all flow cytometry results throughout the manuscript.

- *Fig. 2e-f: is the lowest y-axis tick value 3 or 3.2?*

We have modified the y-axis in this figure to indicate the lowest value as 3.2.

- *Fig. 4j-l: which conditions are compared, is it 0 vs. 12.5 mM? Was a correction for false discovery rate applied?*

We thank the reviewer for this comment. We have revised the figure caption to clarify the ligand conditions that are being compared in this figure and the statistical tests. In addition, the Benjamini-Hochberg adjusted p-values are all reported all flow cytometry results throughout the manuscript.

- *Fig. 4 and the main text: needs consistency in use of eGFP, not GFP.*

We thank the reviewer for this comment. We have revised Figure 4 and the main text to consistently refer to eGFP.

Reviewer #3 (Remarks to the Author):

The authors present a well-written manuscript describing a substantial body of work to develop a method for indirectly assessing the activity of engineered riboswitch designs expressed in mammalian cells that will certainly be of interest to the field. I am particularly impressed by their extensive validation using control ribozymes and libraries. In general, the authors claims are supported by the data presented and the methods are described in sufficient detail for others to reproduce their work. I would support publication in Nature Communications provided that the authors can address the comments below.

This work builds upon existing methods previously described by the Smolke lab and, as the authors acknowledge, the Yokobayashi lab (UC Davis). As in previous methods, RNA-seq is used to quantify levels of transcripts containing self-cleaving aptazyme designs, and the novelty here is the speed of the method and the use of DNA sequence information to normalize RNA data in order to reduce errors arising from inter alia differences in plasmid copy numbers and/or differences in amplification during the workup.

An overall comment is that whilst the method indeed appears to be sound for identifying improved designs present in the libraries, the dynamic range of the best variants is still relatively modest; no ribozyme described here has a dynamic range greater than one order of magnitude. Could the authors comment on whether they feel that this is limited by the designs investigated, or these particular aptamers / ribozyme, or other factors? The method is restricted to very small library sizes (compared with, for example, SELEX libraries) - given this I would suspect that the method may be limited to fine-tuning new designs rather than an extensive exploration of sequence space that may contain more sophisticated aptazymes.

We thank the reviewer for their supportive remarks and suggestions. We feel that the engineered ribozymes in this manuscript not achieving a dynamic range greater than one order magnitude, albeit close, is likely limited by the designs we investigated. To facilitate method development, we chose to focus our efforts on the most well-studied class of ribozymes employed in RNA switch engineering, i.e. the hammerhead ribozymes. Specifically, the sTRSV hammerhead ribozyme was used, which has a basal expression of ~5% of a non-cleaving ribozyme control. While in theory a ribozyme switch could be engineered to have 5% basal expression, sequence constraints posed by the grafted aptamer resulted in many of the highest fold induction switches having basal expressions around 10% to 12%, which limits the maximum possible activation ratios at 100% induction to approximately 10 fold. In addition, ligand permeability issues likely prevent the switches from reaching full induction. Work by others in the field have investigated newer ribozymes, such as the twister ribozyme, twister sister, and pistol ribozymes, which may have lower basal expression levels than sTRSV. Our method is applicable to the engineering ribozyme switches based on these newer ribozyme classes to potentially achieve dynamic ranges greater than one order of magnitude in mammalian cells.

We acknowledge that the method is restricted to small library sizes compare to SELEX libraries, as cell-based assays are inherently limited by plasmid transformation and transfection efficiencies on the order of 10^6 . Therefore, the method cannot select aptamers or sophisticated ribozyme switch designs. However, we would like to clarify that SELEX is an evolution selection process that does not provide activity measurements of all library members simultaneously, which is what our RNA-Seq and FACS-Seq based activity assays can achieve. The library sizes that we employed are a result of our experience in working with this class of ribozyme switches and our understanding of a N5 to N6 loop II sequence being sufficient for identifying optimal sequence variants. Our library design enables us to perform the RNA-Seq sequencing necessary for an N5-6 library for a single ligand on a MiSeq Illumina sequencing platform. However, we believe our method can be used to screen at least an order of magnitude larger libraries to perform extensive exploration of sequence space that contain more sophisticated ribozyme switch library. We estimate that this can be directly scaled up about 15 fold from the 6-well plate well transfection scale in our study to 15 cm dish formats for exploring larger sequence spaces.

We added a discussion to the main text to address the size of the sequence search space as being capable of focused exploration of sequence space on specific library designs with existing aptamers

Specific comments:

1. The authors state that a similar method described by the Yokobayashi lab (Nomura et al. 2017 Chem. Comm.) is limited to screening ribozyme designs of the same length (page 5, line 106). However, as I understand it, whilst the different aptazyme library designs differ in length, within each library the variants examined here are also of the same length. Have the authors examined whether their method indeed allows a library containing length variants (i.e. with indels) to be screened?

We thank the reviewer for this comment. While we did not mix different ribozyme switch libraries during our experiments, the libraries that we have run included loop lengths N5 and N6 (differing by 1 bp in length), and in all libraries, five control ribozymes were spiked in, which are shorter than all the aptamer-integrated libraries by ~15-75 nucleotides. The relative RNA read counts were fit to these 5 control ribozymes so that different libraries sequenced could be scaled to this standard set of 5 control ribozymes and subsequently compared to each other on a standard scale. We believe that running the shorter control ribozymes in each ribozyme switch library supports that our method can be applied to differing length ribozyme switches; however, we have removed the language about the limits of the Yokobayashi lab method to avoid confusion that may arise if interpreted to mean that we mixed different ribozyme switch libraries during our experiments.

2. The authors state that DNA plasmids were extracted from separately transfected cells than those used for the RNA-Seq workup. As transfection efficiency can vary been batches of cells, does this not

unnecessarily introduce a source of error? Would it not have been better to transfect twice as many cells then use half for DNA and half for RNA?

We thank the reviewer for this comment. We did perform the experiments in the way described by the reviewer, i.e., transfected twice as many cells and used half for DNA and half for RNA extraction. We acknowledge that our original description was unclear, and have modified the manuscript text to clarify this point. In addition, we have revised Figure 1c to better reflect this approach.

3. In figure 2d, the change in mCherry fluorescence is used to report activation of a range of theophylline-binding riboswitch variants identified using the RNA-Seq screen. The level of activation at 1mM and 5mM appears to be very similar, indicating that maximum activation is already achieved at 1mM. It would have been reassuring that the switches are working as intended to see a dose response with activation scaling with theophylline concentration. This would be particularly relevant if, for example, a riboswitch is being engineered to provide a quantitative response to a ligand of interest. Do the authors have data for lower concentrations of theophylline? If so do these indeed show a dose response?

We thank the reviewer for this suggestion. We have included new data in the revised manuscript text that provides a dose response curve for a theophylline-responsive switch (new Figure 2e). In addition, we have revised the manuscript text to refer to this new data, and include a new source data file for this set of results.

4. Figure 3b is not referred to in the main text. Presumably a reference to this panel should appear on Page 11 line 253.

We thank the reviewer for this suggestion, and have included a reference to Figure 3b in the indicated location in the text.

5. Page 15. The authors describe how the folinic acid riboswitch library was incompatible with the coding sequence of mCherry ostensibly due to complementarity, leading to misfolding and inactivation of the ribozyme in this library, and hence high basal expression of the reporter. However, in supplementary note 3 (and referring to data in supplementary figure 8) the authors note that bulk cleavage of this library was comparable to other libraries in vitro, which would not be the case if misfolding occurred in this assay. Presumably the mCherry RNA sequence alone is insufficient to cause misfolding (as the authors claim in the main text, page 15 lines 345-8), and other factors affect the folding inside cells – this should be noted in the main text.

We thank the reviewer for this comment. The *in vitro* cleavage assays did not include the mCherry sequence, but rather were performed with the RNA encoding the ribozyme switch and A-rich insulator flanking sequences. We do not have direct experimental data that demonstrate that the folinic acid aptamer is misfolding due to the mCherry sequence. The sequence interactions between the folinic acid aptamer and mCherry leading to misfolding were predicted by analyzing the sequence via an RNA secondary structure folding program. In addition, the putative interaction between the aptamer and mCherry sequence is indirectly supported by our observations that changing the mCherry sequence to an eGFP sequence recovered the expected low reporter gene expression level, presumably resulting from recovering cleavage activity of the ribozyme switch in that sequence context. We have revised the manuscript text to clarify these points in the Discussion section and Supplementary Information.

6. I found the section dealing with conservation in sequence and structure of ribozymes quite difficult to follow. In particular, given that (as the authors acknowledge on page 19 lines 438-441) interactions between the loop residues varied in the libraries and the linked aptamers is likely to be

specific for the aptamers (and the ribozyme) used in this study, I am unsure how the authors' findings "could be used to inform future rational design efforts" (Abstract lines 30-32) in any concrete way for other aptamers and ribozymes – could the authors be more specific in this claim?

We thank the reviewer for this comment. We recognize that with the data in the current manuscript, rational design is not possible; but with more examples of functional switches in the future, we would be able to start inferring the size and complexity of this sequence-function landscape. For a stop gap semi-rational approach right now, we proposed sequentially screening 8 designs with aptamers on loop I and [A|C|T|G]RRA[A|G] on loop II first to determine the best base identity at the first and last position, before screening the remaining 4 sequences in the middle of the loop to obtain a near optimal switch, as indicated in Figure 5e. We corrected the abstract to reflect that our current datasets allow us to make semi-rational designs, i.e., design around 8-12 sequences for any of the four aptamers, and the same design principles here and described in the section "Conserved sequence and structural motifs underlie highly functional ribozyme switches" may possibly be applied to other aptamers, to engineer switches that can display large dynamic range, while the RNA-Seq method has the potential to generate larger datasets to eventually inform rational design.

7. As the authors acknowledge, the FACS-Seq data appears to be very noisy. I am unfamiliar with the method used to improve the signal to noise so cannot comment on whether this is appropriate.

We thank the reviewer for this comment. We would like to further clarify that we applied denoising concepts in current machine learning literature, commonly found in image processing. Similar strategies employed in denoising RNA-Seq inspired us to leverage the similarities in sequences in the data representing sequence-function relationships, for reducing the noise in the FACS-Seq and RNA-Seq data, to generate "corrected" switch activity values. We now include a few citations to the relevant literature in the main text and supplementary note to provide more context to the method.

8. In Supplementary figure 5, some of the plots appear to show two fit lines –are these a fit of each replicate data set, or of the raw vs denoised data for both replicates? Why do some panels only show one fit?

We thank the reviewer for this comment. In each of the plots there are two fit lines shown, one for the fit between replicates for raw data and another for the denoised data in order to generate the R^2 correlation. In the panels where there seems to be only one fit, there are actually two lines directly overlapping, suggesting better agreement between the raw and denoised data. We have modified the figure caption to clarify these points.

9. In Supplementary figure 8, the legend suggests that the pairs of reactions shown in the gel(s) are -/+ ligand – this should be indicated on the figure. The reaction conditions and timing should be indicated in the figure legend.

We thank the reviewer for these suggestions. We have revised Supplementary Figure 8 such that -/+ ligand are indicated on the figure and the reaction conditions and timing are indicated in the figure caption.

References

1. Zhong, G., Wang, H., Bailey, C. C., Gao, G. & Farzan, M. Rational design of aptazyme riboswitches for efficient control of gene expression in mammalian cells. *Elife* **5**, (2016).
2. Nomura, Y., Zhou, L., Miu, A. & Yokobayashi, Y. Controlling mammalian gene expression by allosteric hepatitis delta virus ribozymes. *ACS Synth. Biol.* **2**, 684–689 (2013).
3. Stifel, J., Spöring, M. & Hartig, J. S. Expanding the toolbox of synthetic riboswitches with guanine-dependent aptazymes. *Synthetic Biology* **4**, (2019).
4. Felletti, M., Klauser, B. & Hartig, J. S. Screening of Genetic Switches Based on the Twister Ribozyme Motif. *Methods in Molecular Biology* 225–239 (2016). doi:10.1007/978-1-4939-3197-2_19
5. Beilstein, K., Wittmann, A., Grez, M. & Suess, B. Conditional control of mammalian gene expression by tetracycline-dependent hammerhead ribozymes. *ACS Synth. Biol.* **4**, 526–534 (2014).
6. Chen, Y. Y., Jensen, M. C. & Smolke, C. D. Genetic control of mammalian T-cell proliferation with synthetic RNA regulatory systems. *Proceedings of the National Academy of Sciences* **107**, 8531–8536 (2010).
7. Bell, C. L. *et al.* Control of alphavirus-based gene expression using engineered riboswitches. *Virology* **483**, 302–311 (2015).
8. Wong, R. S., Chen, Y. Y. & Smolke, C. D. Regulation of T cell proliferation with drug-responsive microRNA switches. *Nucleic Acids Res.* **46**, 1541–1552 (2018).
9. Gu, H., Furukawa, K. & Breaker, R. R. Engineered Allosteric Ribozymes That Sense the Bacterial Second Messenger Cyclic Diguanosyl 5'-Monophosphate. *Analytical Chemistry* **84**, 4935–4941 (2012).
10. Win, M. N. & Smolke, C. D. A modular and extensible RNA-based gene-regulatory platform for engineering cellular function. *Proceedings of the National Academy of Sciences* **104**, 14283–14288 (2007).
11. McKeague, M., Wang, Y.-H. & Smolke, C. D. In Vitro Screening and in Silico Modeling of RNA-Based Gene Expression Control. *ACS Chem. Biol.* **10**, 2463–2467 (2015).
12. Nguyen Quang, N., Perret, G. & Ducongé, F. Applications of High-Throughput Sequencing for In Vitro Selection and Characterization of Aptamers. *Pharmaceuticals* **9**, (2016).
13. Mahen, E. M., Harger, J. W., Calderon, E. M. & Fedor, M. J. Kinetics and thermodynamics make different contributions to RNA folding in vitro and in yeast. *Mol. Cell* **19**, 27–37 (2005).

REVIEWERS' COMMENTS:

Reviewer #2 (Remarks to the Author):

The authors have suitably addressed all comments I raised in the previous round of review. To my reading, the authors also appear to have made a good faith attempt to respond to comments raised by my fellow reviewers. I am compelled by the potential impact of the technical advance made here, even if the specific riboswitches described may require coupling with downstream signal processing to achieve the application-specific combination of ligand concentration and fold-induction required; these are points raised by Reviewer 1 and which the authors address in their rebuttal. Although a comprehensive discussion of this point is beyond the scope of this manuscript, it could be helpful to the reader to add a brief mention of this concept, since it provides some important context in which to interpret the slippery question of “how great must the fold-induction be to be useful?” (for which there is no one answer, to be clear).

Reviewer #3 (Remarks to the Author):

(As noted by the other reviewers), the modest dynamic range of the described switches reduces the impact of this paper, and (as the authors generally acknowledge in their responses) although these type of approach has potential, it has not yet been used to develop significantly improved riboswitch designs beyond those already described in the literature. The resubmitted manuscript does not overturn this criticism; nonetheless, I believe thorough, good quality investigations taking steps in this direction, as this manuscript describes, will be of interest to the field and as such I continue to support publication in Nature Communications, provided that the authors address some minor issues, described below.

Response to specific comments (Reviewer #3):

1. I am satisfied with the changes. No further comments.
2. I am unsure where the authors have clarified this point – a sentence describing the merits of quantifying DNA post-transfection has been inserted on page 7, but I cannot see any mention of this being from the same cell pool used for RNA-Seq. The authors updated Figure 1c appears to now show the other scenario, two separate batches of cells, implied by two black lines / arrows.
3. The new data does indeed show a theophylline dose response curve for three of the riboswitches, albeit within the modest dynamic range discussed elsewhere. No further comments.
4. No further comments.
5. If the authors have not measured the aptamer misfolding directly, and have used a secondary structure prediction program to assess the likelihood of interference with folding of the aptamer, this should be noted in the discussion, and the software acknowledged. E.g. on page 22, The line “... which may be due to the folinic acid aptamer sequence forming undesirable base pair interactions with sequences in the mCherry transcript.” could be modified to “... which may be due to the folinic acid aptamer sequence forming undesirable base pair interactions with sequences in the mCherry transcript, as suggested by secondary structure prediction using (software name)”.
6. The authors have satisfactorily clarified this point.

7. No further comments.

8. The fits in this figure could be clarified by using dashed lines of different colours for the fits for raw and denoised data.

9. The reaction conditions have been added to the figure legend but $-/+$ ligand still do not appear to be indicated on the figure.

RESPONSE TO REVIEWER COMMENTS

Reviewer #2 (Remarks to the Author):

The authors have suitably addressed all comments I raised in the previous round of review. To my reading, the authors also appear to have made a good faith attempt to respond to comments raised by my fellow reviewers. I am compelled by the potential impact of the technical advance made here, even if the specific riboswitches described may require coupling with downstream signal processing to achieve the application-specific combination of ligand concentration and fold-induction required; these are points raised by Reviewer 1 and which the authors address in their rebuttal. Although a comprehensive discussion of this point is beyond the scope of this manuscript, it could be helpful to the reader to add a brief mention of this concept, since it provides some important context in which to interpret the slippery question of “how great must the fold-induction be to be useful?” (for which there is no one answer, to be clear).

We thank the reviewer for the encouraging remarks and suggestion. We have revised the text to include a statement in the discussion that exactly how great the fold-induction needs to be in order to be useful is application dependent and the specific RNA switches we engineered may require coupling with downstream signal processing elements to achieve the application-specific combination of ligand concentration and fold-induction required.

Reviewer #3 (Remarks to the Author):

(As noted by the other reviewers), the modest dynamic range of the described switches reduces the impact of this paper, and (as the authors generally acknowledge in their responses) although these type of approach has potential, it has not yet been used to develop significantly improved riboswitch designs beyond those already described in the literature. The resubmitted manuscript does not overturn this criticism; nonetheless, I believe thorough, good quality investigations taking steps in this direction, as this manuscript describes, will be of interest to the field and as such I continue to support publication in Nature Communications, provided that the authors address some minor issues, described below.

We thank the reviewer for the encouraging remarks and constructive suggestions. We also thank the reviewer for confirming that we adequately addressed points 1, 3, 4, 6 and 7 below.

Response to specific comments (Reviewer #3):

1. I am satisfied with the changes. No further comments.

2. I am unsure where the authors have clarified this point – a sentence describing the merits of quantifying DNA post-transfection has been inserted on page 7, but I cannot see any mention of this being from the same cell pool used for RNA-Seq. The authors updated Figure 1c appears to now show the other scenario, two separate batches of cells, implied by two black lines / arrows.

We thank the reviewer for this comment. We have modified the text to explicitly mention that the DNA is extracted from a separately transfected cell pool.

3. The new data does indeed show a theophylline dose response curve for three of the riboswitches, albeit within the modest dynamic range discussed elsewhere. No further comments.

4. No further comments.

5. If the authors have not measured the aptamer misfolding directly, and have used a secondary structure prediction program to assess the likelihood of interference with folding of the aptamer, this should be noted in the discussion, and the software acknowledged. E.g. on page 22, The line “... which may be due to the folinic acid aptamer sequence forming undesirable base pair interactions with sequences in the mCherry transcript.” could be modified to “... which may be due to the folinic acid aptamer sequence forming undesirable base pair interactions with sequences in the mCherry transcript, as suggested by secondary structure prediction using (software name)”.

We thank the reviewer for this suggestion. We have modified the text to explicitly mention the software used (i.e., RNAstructure).

6. The authors have satisfactorily clarified this point.

7. No further comments.

8. The fits in this figure could be clarified by using dashed lines of different colours for the fits for raw and denoised data.

We thank the reviewer for the suggestion and fixed the dashed lines to use different colors, dash style, and thickness to better distinguish between raw and denoised data.

9. The reaction conditions have been added to the figure legend but -/+ ligand still do not appear to be indicated on the figure.

We apologize that we inadvertently left out the -/+ ligand annotation, but now have included it in the revised figure.